# Towards Robust Multi-Modal Reasoning via Model Selection

**Xiangyan Liu**[3,*]  **Rongxue Li**[2,1,*]  **Wei Ji**[3]  **Tao Lin**[1,†]
`liu.xiangyan@u.nus.edu; lirongxue@westlake.edu.cn;`
`weiji0523@gmail.com; lintao@westlake.edu.cn`
[1]Westlake University  [2]Zhejiang University  [3]National University of Singapore

## Abstract

The reasoning capabilities of LLM (Large Language Model) are widely acknowledged in recent research, inspiring studies on tool learning and autonomous agents. LLM serves as the "brain" of the agent, orchestrating multiple tools for collaborative multi-step task solving. Unlike methods invoking tools like calculators or weather APIs for straightforward tasks, multi-modal agents excel by integrating diverse AI models for complex challenges. However, current multi-modal agents neglect the significance of model selection: they primarily focus on the planning and execution phases, and will mainly invoke predefined task-specific models for each subtask, making the execution fragile. Meanwhile, other traditional model selection methods are either incompatible with or suboptimal for the multi-modal agent scenarios, due to ignorance of dependencies among subtasks arising by multi-step reasoning.

To this end, we identify the key challenges therein and propose the $M^3$ framework as a plug-in with negligible runtime overhead at test-time. This framework improves model selection and bolsters the robustness of multi-modal agents in multi-step reasoning. In the absence of suitable benchmarks, we create MS-GQA, a new dataset specifically designed to investigate the model selection challenge in multi-modal agents. Our experiments reveal that our framework enables dynamic model selection, considering both user inputs and subtask dependencies, thereby robustifying the overall reasoning process. Our code and benchmark: [https://github.com/LINs-lab/M3](https://github.com/LINs-lab/M3).

## 1 Introduction

Large Language Models (LLMs) (Brown et al., 2020; Ouyang et al., 2022; Chowdhery et al., 2022; Zhang et al., 2022b; Touvron et al., 2023) recently emerged to show great potential for achieving human-level intelligence, leveraging the key reasoning ability to tackle complex problems.

As a key step towards artificial general intelligence, the study on multi-modal learning has soon evolved into two paradigms, either training large end-to-end models like PaLM-E (Driess et al., 2023) and Mini-GPT4 (Zhu et al., 2023) for direct task resolution, or employing LLMs to decompose tasks into subtasks for smaller yet specific models (Gupta & Kembhavi, 2023; Surís et al., 2023; Shen et al., 2023; Gao et al., 2023a). The latter paradigm—as evidenced in the significant attention since tool learning (Schick et al., 2023) and autonomous agents (Reworkd, 2023; Richards et al., 2023)—demonstrates the immense potential in addressing complex real-world problems, where multiple AI models collaborate through *multi-modal multi-step reasoning process*.

In the realm of multi-modal reasoning scenarios, existing multi-modal agents (Gupta & Kembhavi, 2023; Shen et al., 2023) emphasize planning and execution phases, while neglecting the critical model selection phase. As exemplified in Figure 1 (a & b), a simplistic model selector relies on predefined task-specific models for subtasks, increasing the likelihood of intermediate errors and compromising the overall reasoning process. Moreover, existing traditional model selection methods, though effective in various domains (Zhao et al., 2021; Park et al., 2022; Lee et al., 2022; Zitovsky et al., 2023), primarily focus on selecting a single model from multiple candidates per sample.

---

*Equal contribution. Work was done during Xiangyan's visit to Westlake University.
†Corresponding author.

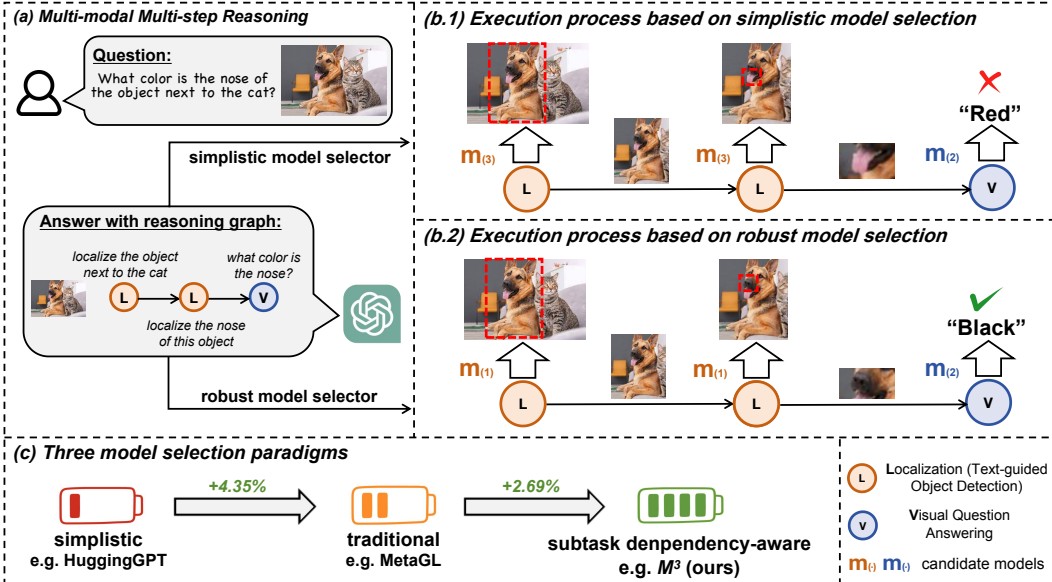

Figure 1: **Illustration of the multi-modal multi-step reasoning process and three model selection paradigms within**. *(a)* shows how multi-modal agents utilize LLMs to decompose complex multi-modal tasks, resulting in a multi-step reasoning process where each node corresponds to a simpler yet more specific subtask. *(b)* highlights that compared to a robust model selector, simplistic model selection methods are more prone to generating wrong outcomes at intermediate subtask stages, thereby impacting the ultimate reasoning result. Here, $m_{(i)}$ indicates the $i$-th model is selected and the color means the corresponding subtask type. *(c)* numerically illustrates the comparative outcomes of model selection methods from different paradigms in multi-modal reasoning.

Adapting these methods to multi-modal reasoning scenarios, which necessitate multiple models for subtasks, is challenging due to the oversight of subtask dependencies.

To this end, we formally define the problem of model selection in multi-modal reasoning scenarios as our first contribution, and then introduce the $M^3$ framework (**M**odel Selector for **M**ulti-**M**odal Reasoning) as our preliminary remedy for the field. Given the lack of benchmarks, we further create a new dataset named MS-GQA (Model Selection in GQA (Hudson & Manning, 2019)) to facilitate research in the field. In detail, $M^3$ represents the multi-step reasoning process as a computation graph, with nodes corresponding to reasoning subtasks. Multi-modal encoders and models' embedding table transform input and selected models into node features, where a computation graph learner then models the relationships between input, selected models, and subtask dependencies to predict the execution status. $M^3$ showcases superior performance in the model selection process for multi-modal models, with trivial overhead in selection.

**Our key contributions are summarized as follows:**

- We formulate the model selection problem in multi-modal reasoning contexts as an initial endeavor.
- We introduce $M^3$, a model selection framework for multi-modal models with multi-step reasoning, jointly modeling the relationship between samples, selected models, and subtask dependencies.
- We create a comprehensive dataset called MS-GQA to facilitate the research for the community.
- We provide an effective yet efficient model selection solution, with trivial test-time overhead.

## 2 RELATED WORK

### 2.1 MULTI-STEP REASONING WITH LLM

A line of work utilizes LLMs to interact with simple APIs, including WebGPT (Nakano et al., 2021), Toolformer (Schick et al., 2023), PAL (Gao et al., 2023b), and ToolkenGPT (Hao et al., 2023), enabling the manipulation of tools like web browser and calculators. Their capabilities are enhanced by behaving as autonomous agents (Richards et al., 2023; Nakajima, 2023; Reworkd, 2023), where they reason, break down complex tasks into subtasks, and iteratively execute those subtasks until the desired goals are achieved. Note that these studies only call simple deterministic APIs for text tasks. To tackle complex multi-modal tasks, researchers are increasingly expanding tool libraries with trained AI models (Wu et al., 2023; Huang et al., 2023; Surís et al., 2023). For instance, Vis-Prog (Gupta & Kembhavi, 2023) and HuggingGPT (Shen et al., 2023) employ LLMs to decompose complex tasks into subtasks and connect various AI models to address them. Chameleon (Lu et al.,

2023) is a plug-and-play compositional reasoning framework using a richer set of tools. Unlike the aforementioned approaches that provide a static plan without considering dependent subtasks, AssistGPT (Gao et al., 2023a) and AVIS (Hu et al., 2023) dynamically strategize the utilization of external tools based on the intermediate results of multi-step reasoning. However, all existing methods lack model selection consideration, resulting in reasoning instability.

## 2.2 MODEL SELECTION

The research on model selection can date back to Forster (2000), and was further extensively investigated in various aspects: 1) meta-learning methods (Zhao et al., 2021; 2022; Park et al., 2022), 2) non-meta-learning methods (Ying et al., 2020; Zohar et al., 2023), 3) model selection for ensemble (Kotary et al., 2023), 4) model selection with language models (Zhao et al., 2023; Hari & Thomson, 2023), 5) new metrics for model selection (Zhang et al., 2021; Yang et al., 2023a), and others (Chen et al., 2023; Lee et al., 2022; Zitovsky et al., 2023). We discuss the most relevant two lines of research as follows. Meta-learning-based approaches utilize the similarity between new instances and historical instances to predict the performance of candidate models. For example, MetaOD and ELECT (Zhao et al., 2021; 2022) address the unsupervised outlier model selection problem, where they extract meta-features by considering input-specific characteristics. MetaGL (Park et al., 2022) further extends (Zhao et al., 2021) to select a model for each new graph. However, in our multi-modal multi-step reasoning scenarios, there is currently a lack of established approaches for extracting instance-wise meta-features. In contrast, non-meta-learning-based methods resort to using complex networks to learn the relationship between instances and model choices, without using meta-features. Auto-Selector (Ying et al., 2020) employs a pre-trained model selector and parameter estimator to automatically choose an anomaly detection model for incoming time-series data. LOVM (Zohar et al., 2023) employs textual dataset descriptions to train a linear model that predicts the performance of vision-language candidate models. EMMS (Meng et al., 2023) employs weighted linear regression to estimate multi-modal model transferability. Note that these methods are restricted to one-step selection and overlook subtask dependencies in multi-step reasoning, and thus infeasible for model selection in multi-step reasoning. In the context of using LLMs for multi-modal models in multi-step reasoning, existing methods either rely on external metrics like download counts for model selection, or explicitly specify the use of a specific version of a model for a designated task without any selection process (Surís et al., 2023). None of the prior work takes the challenges of multi-step reasoning dependency into account.

## 3 MODEL SELECTION HARNESSES THE MULTI-MODAL REASONING

### 3.1 ON THE CHALLENGES OF MULTI-MODAL MULTI-STEP REASONING

Given a complicated input query, the contemporary multi-modal multi-step reasoning process normally involves calling LLMs to decompose users' input to subtasks, in which the reasoning logic can be constructed as a task graph by composing dependent intermediate subtasks.

The problem of selecting models for subtasks along the task graph can be defined below:

**Definition 3.1** (Model selection for each subtask type on a multi-modal task graph). *Let $m(i, j, t)$ denote a model choice of subtask type $t$ for the $i$-th sample $\mathbf{x}_i$, where $m(i, j, t)$ indicates the $j$-th choice from the model zoo for subtask type $t$ on $\mathbf{x}_i$. Each subtask type $t$ has $n_t$ choices, forming a set $\{m(\cdot, j, t) \mid j \in [n_t]\}$.*

*Assume a task graph consists of $K$ subtask types, then all potential model selection choices for the task graph can be defined as $\mathcal{C} = \{m(\cdot, j, t) \mid j \in [n_t], t \in [K]\}$, with size $|\mathcal{C}| = \prod_{t=1}^{K} n_t$. The optimal choice of models on the task graph for $i$-th input can be represented as $\mathbf{c}_i^\star := \{m(i, j_t^\star, t) \mid t \in [K]\}$, where we select the optimal model index $j_t^\star$ for each subtask type $t$.*

Though Definition 3.1 fully characterizes the procedure of traditional model selection methods as evidenced in Figure 2 and Section 2.2, it leaves the unique challenge emerged in the multi-modal multi-step reasoning untouched, namely the critical *subtask dependency* defined below.

**Definition 3.2** (Subtask dependency on a multi-modal task graph). *Given the $i$-th input $\mathbf{x}_i$ with embeddings from various modalities. Its multi-step reasoning procedure decomposed by an LLM can be described as a directed acyclic computation graph $\mathcal{G}_i = \{\mathcal{V}_i, \mathcal{T}_i, \mathcal{E}_i\}$ with nodes corresponding to multi-modal subtasks, where*

- *$\mathcal{V}_i = \{v_{i,k} \mid k \in [L]\}$ is the subtask nodes in the graph, $v_{i,k}$ is the $k$-th node in $\mathcal{G}_i$, and $L := |\mathcal{V}_i|$;*
- *$\mathcal{T}_i = \{t_{i,k} \mid k \in [L]\}$ is the set of subtask types in $\mathcal{V}_i$, $t_{i,k}$ is the subtask type of $k$-th node in $\mathcal{G}_i$;*
- *$\mathcal{E}_i$ is the set of edges that connect pairs of subtask nodes in the graph $\mathcal{G}_i$.*

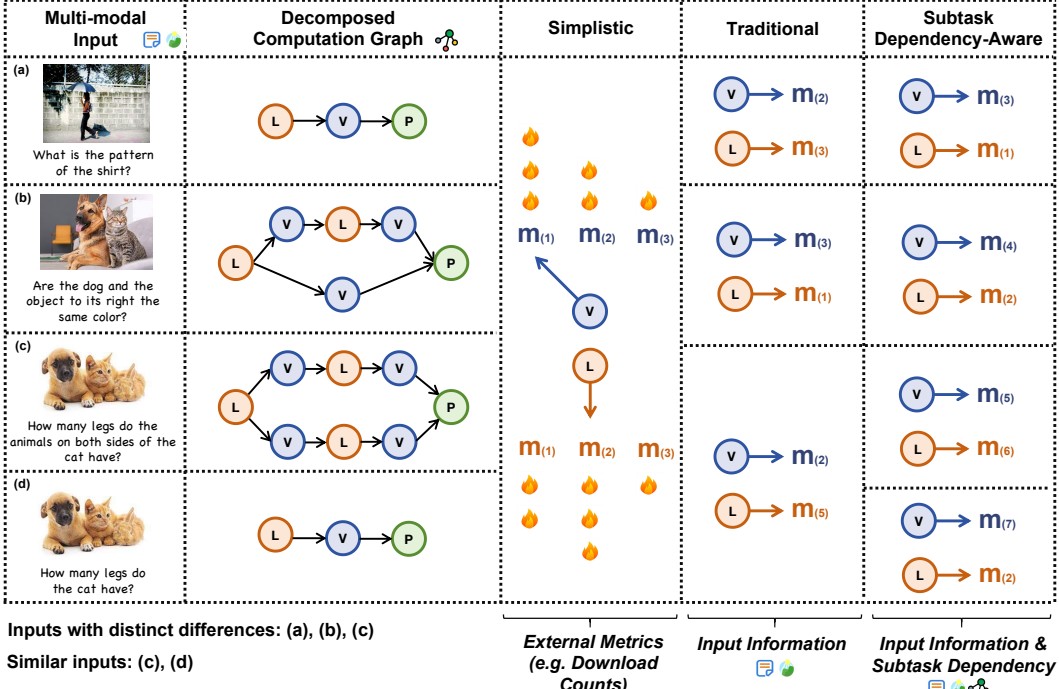

Figure 2: **Comparison of three model selection paradigms under various inputs.** The model selection processes of the three paradigms, from left (simplistic) to right (subtask dependency-aware), become progressively more fine-grained. "Simplistic" is inflexible and can be considered input-agnostic. "Traditional" can solely depend on subtask type and the corresponding original input information for model selection. When inputs are similar, "Traditional" cannot provide as diverse model selections as "Subtask Dependency-Aware", which leverages differences in reasoning logic to offer more varied and suitable model choices. Note that node P (green circle) in the figure denotes Python module invocation, which does not entail model selection.

**Remark 3.3** (Generalizing model selection beyond the subtask type). *To ease the presentation, Definition 3.1, Definition 3.2, and the main content below only consider selecting the model per subtask type. Though being sufficient for traditional model selection methods, such formulation can be further generalized to performing model selection per subtask node (rather than per subtask type) in our case, and we leave it for future research.*

**Challenges: on the infeasibility of adapting existing model selection methods.** Existing methods (see details in our Section 2.2), either the current preliminary strategies for multi-modal agents (Gupta & Kembhavi, 2023; Surís et al., 2023; Shen et al., 2023; Gao et al., 2023a), or model selection methods from other domains (Zhao et al., 2021; 2022; Park et al., 2022; Ying et al., 2020; Zohar et al., 2023), do not take the subtask dependency into account—as evidenced in the following case study—making the model selection for multi-modal model with multi-step reasoning non-trivial. We describe the previous two model selection paradigms (see examples in Figure 2):

1. This paradigm is primarily applied in some recent multi-modal agent frameworks, where each subtask type $t$ will be straightforwardly allocated to a model via some external metrics (e.g., download counts or recent release dates).
2. As has been widely used in other fields like outlier detection and graph learning, this paradigm focuses on matching the optimal model through the input information as well as the subtask type.

Given the limitations of previous methods in adapting to the timely multi-modal reasoning scenarios, in the following section, we incorporate the graph information $\mathcal{G}_i$ to capture subtask dependencies.

### 3.2 $M^3$: A FRAMEWORK OF MODEL SELECTION FOR MULTI-MODAL REASONING

An ideal model selection solution on multi-modal reasoning scenarios discussed above motivates us to jointly model the subtask dependency with input sample features. We depict our unified model selection framework $M^3$ below towards this goal.

**Overview.** The model selection process can be viewed as estimating the performance of a multi-modal input over the corresponding task graph, in which the choice of model selection at each subtask

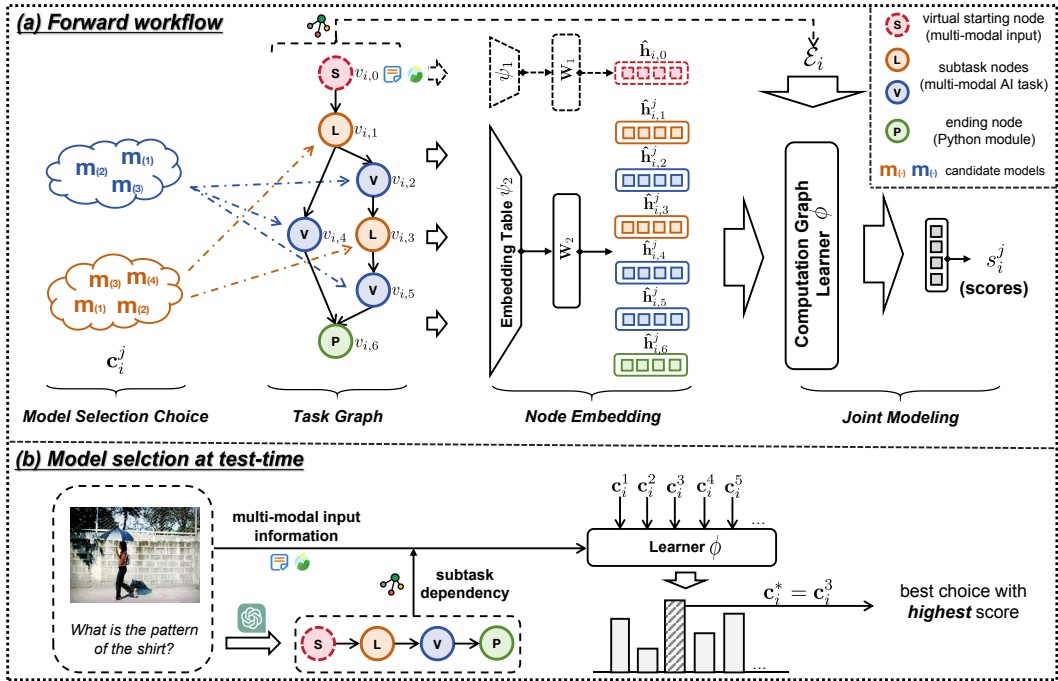

Figure 3: **Illustration of $M^3$:** **(a)** depicts the forward computation process: 1) **Task Graph:** An initial virtual node represents the multi-modal input. Specific models are assigned to each subtask node based on the respective subtask type. 2) **Node Embedding:** Features are extracted using the multi-modal encoder $\psi_1(\cdot)$ and embedding table $\psi_2(\cdot)$ for the initial virtual node and subtask nodes. 3) **Computation Graph Learner:** The computation graph, including node features and subtask dependencies (edges $\mathcal{E}_i$), serves as input to learner $\phi(\cdot)$, contributing to the predicted execution status $s_i^j$. **(b)** illustrates the process of ranking and selecting the model selection choice with a greater likelihood of success.

node propagates to the next node on the directed task graph. We introduce the notion of meta-training and train a proxy on it to suggest the optimal model choice of task graph on unseen input. In detail,

- **Training the proxy.** Given an input sample $\mathbf{x}_i \in \mathcal{X}_{\text{train}} := \{\mathbf{x}_1, \ldots, \mathbf{x}_N\}$, for each choice of models on the task graph $\mathbf{c}_i^j \in \mathcal{C}$, we aim to model the relationship $\phi \circ \psi$ between $(\mathbf{x}_i, \mathcal{G}_i, \mathbf{c}_i^j)$ and the execution status $p_i^j \in \{0, 1\}$, where $\phi$ and $\psi$ denote the learner and feature extractor, respectively. Here we simplify our setting by only considering the binary execution status, but the principles therein can be generalized to the continuous case.
- **Model selection for task graph on the unseen sample.** We estimate the status of potential model choices $\{s_i^j := \phi \circ \psi(\mathbf{x}_i, \mathcal{G}_i, \mathbf{c}_i^j) \mid \mathbf{c}_i^j \in \mathcal{C}\}$, and only keep executable choices for further selection.

### 3.2.1 TRAINING THE PROXY

**Lookup table and node embedding $\psi$.** We adopt a similar approach to treatments in multi-modal learning and GNNs (Li et al., 2022; Veličković et al., 2017). Using $\psi := [\psi_1, \psi_2]$—where $\psi_1$ denotes the multi-modal encoder and $\psi_2$ represents the model embedding table—we map subtask nodes to their corresponding embeddings through $\boldsymbol{H}_i^j := \psi(\mathbf{x}_i, \mathcal{G}_i, \mathbf{c}_i^j) \in \mathbb{R}^{(L+1) \times d}$.

More specifically, we first slightly abuse the notation and form an augmented computation graph $\mathcal{G}_i = (\mathcal{V}_i, \mathcal{T}_i, \mathcal{E}_i)$: we introduce a virtual starting node $v_{i,0}$ to incorporate the multi-modal input information $\mathbf{x}_i$ and use subtask nodes $\{v_{i,k} \mid k \in [L]\}$ to indicate the execution dependency of subtasks. We then employ two encoders $\psi_1, \psi_2$ separately to retrieve the node embeddings and use additional $\boldsymbol{W}_1 \in \mathbb{R}^{d_1 \times d}$ and $\boldsymbol{W}_2 \in \mathbb{R}^{d_2 \times d}$ to unify them in a shared feature space.

- For the virtual starting node $v_{i,0}$, we utilize the off-the-shelf multi-modal encoder to extract node embedding as $\mathbf{h}_{i,0} = \psi_1(\mathbf{x}_i) \in \mathbb{R}^{d_1}$, where $d_1$ is the input embedding dimension.
- For other nodes, we use a lookup table to retrieve node embedding for each subtask type, namely $\mathbf{h}_{i,k}^j = \psi_2\left(m(i, j, t_{i,k})\right) \in \mathbb{R}^{d_2}$, where $t_{i,k} \in \mathcal{T}_i$ and $d_2$ is the model embedding dimension.

**Joint modeling of node embeddings and subtask dependency $\phi$.** We further leverage a *computation graph learner* to learn the task embedding over 1) sample embeddings, 2) model embeddings, and 3) subtask dependency information. A final linear layer with a non-linear activation function is stacked

on top of the learned task embedding to estimate $s_i^j := \phi \circ \psi(\mathbf{x}_i, \mathcal{G}_i, \mathbf{c}_i^j)$, where $\mathbf{c}_i^j \in \mathcal{C}$. In detail,

$$\mathbf{H}_i^j = \psi(\mathbf{x}_i, \mathcal{G}_i, \mathbf{c}_i^j), \qquad s_i^j = \phi(\mathbf{H}_i^j, \mathcal{E}_i). \tag{1}$$

The output $s_i^j$ measures the degree of match between input sample $\mathbf{x}_i$ and model selection choice $\mathbf{c}_i^j$, where a higher value signifies a greater likelihood of success in a multi-modal reasoning scenario. Note that theoretically, any neural network capable of handling directed acyclic graphs can serve as the backbone for a computation graph learner. See Section 4.2 and Appendix D.3 for details.

**Optimization over $\psi$ and $\phi$.** For a given input sample $\mathbf{x}_i$, we aim to learn the model to estimate execution status per the choice $\mathbf{c}_i^j$ of models along the task graph, namely correlating $s_i^j \in [0, 1]$ with ground truth $p_i^j \in \{0, 1\}$. Thus, the optimization can be viewed as a multi-label classification problem. The conventional choice of using instance-wise Binary Cross-Entropy (BCE) loss in the multi-label classification community (Tsochantaridis et al., 2005; Wehrmann et al., 2018) only aims to build a mapping between $(\mathbf{x}_i, \mathcal{G}_i, \mathbf{c}_i^j)$ and $s_i^j$, and thus suffers from the optimization difficulty caused by prediction independency of $\mathbf{c}_i^j \in \mathcal{C}$ on the input $\mathbf{x}_i$.
Therefore, we employ Categorical Cross-Entropy (CCE) (Su et al., 2022) as our objective function, using a list-wise approach to model $\psi$ and $\phi$ for $(\mathbf{x}_i, \mathcal{G}_i, \mathcal{C})$ and $\{s_i^j \mid \mathbf{c}_i^j \in \mathcal{C}\}$. Our goal is to promote higher scores $s_i^j$ for executable choices ($p_i^j = 1$) and lower scores for non-executable choices:

$$\mathcal{L}_i = \log\left(1 + \sum_{\mathbf{c}_i^j \in \mathcal{C}} 1_{p_i^j = 0} \exp(s_i^j)\right) + \log\left(1 + \sum_{\mathbf{c}_i^j \in \mathcal{C}} 1_{p_i^j = 1} \exp(-s_i^j)\right). \tag{2}$$

See Appendix B and D.4 for more details on loss design and a comparison of different loss functions.

### 3.2.2 MODEL SELECTION FOR TASK GRAPH ON THE UNSEEN SAMPLE

Once we learn the relationship $\phi \circ \psi$, we can estimate execution status $s_i^j$ for all choices of models on the task graph $\mathbf{c}_i^j \in \mathcal{C}$, and transfer to the final model selection upon other criteria. For the sake of simplicity, in our evaluation we primarily select the $\mathbf{c}_i^\star$ via the maximal execution probability: $\mathbf{c}_i^\star = \arg\max_{\mathbf{c}_i^j \in \mathcal{C}} \phi \circ \psi(x_i, \mathcal{G}_i, \mathbf{c}_i^j)$. Other metrics, e.g. computation cost, can be further integrated to trade off efficiency and robustness.

## 4 EXPERIMENTS

### 4.1 BENCHMARK: MS-GQA

As our side contribution, we introduce the first benchmark, MS-GQA (Model Selection in GQA (Hudson & Manning, 2019)), to explore the model selection methods on multi-modal reasoning scenarios. MS-GQA primarily evaluates model selection choices of scenarios using VisProg (Gupta & Kembhavi, 2023) as the autonomous agent to solve AI tasks on source dataset GQA. The choice of autonomous agents and datasets can be

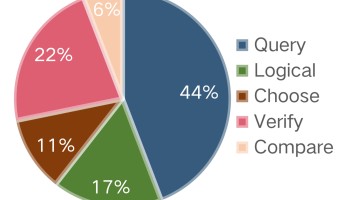

Figure 4: **The proportions of various structural categories in GQA.**

flexibly replaced, such as substituting VisProg with HuggingGPT (Shen et al., 2023) or other related works (Surís et al., 2023; Gao et al., 2023a), and replacing GQA with NLVR2 (Suhr et al., 2018) or even other in-the-wild application scenarios (Surís et al., 2023; Yang et al., 2023c).
Our benchmark considers the structural variations of tasks in GQA, and thus facilitates comprehensive method evaluation and assesses robustness under diverse test distributions. Currently, we introduce 5 task categories, namely *Query, Choose, Compare, Verify,* and *Logical.* The tasks cover 9 functional components (subtask types), of which 7 out of 9 components align with specific Python modules, eliminating the need for model selection. The remaining two components involve "LOC" (localization, text-guided object detection) and "VQA" (visual question answering), offering 10 and 7 candidate models respectively. We have currently collected the binary execution results for 8,426 samples across 70 valid model selection choices. For more details, refer to Appendix A.

### 4.2 EXPERIMENTAL SETTINGS

**Baselines.** As discussed in Section 2.2, the multi-modal model currently falls short of model selection methods. To justify the effectiveness of our solution, we extend a range of representative methods to multi-modal reasoning scenarios as two groups of baseline references illustrated below. ***Training-free*** selects one model for each subtask type based on the prior knowledge or external metrics (e.g., download counts, citations, and publication dates), without considering input information:

- RANDOM: Randomly choose models for each subtask type on the task graph for every sample;
- VISPROG (Gupta & Kembhavi, 2023): Follow the default choice in the original paper of VisProg and select a deterministic candidate model per subtask type;
- EXMETRIC: Incorporate external metrics for model ranking and selection. This baseline can be generalized as a paradigm that utilizes external metrics for model selection; [1]
- GLOBALBEST (Park et al., 2022; Zhao et al., 2021): Select models on the task graph that yield the highest average performance across all training samples, without considering input details.

*Training-based* focuses on leveraging a trainable proxy to find the optimal model. We adapt methods from other domains and form our baselines to meet the requirements of the multi-modal reasoning context. Note that tuning and improving these methods are beyond the scope of this paper.

- NCF (He et al., 2017): A representative model selection approach using collaborative filtering. It uses a neural network to model the interaction between samples and models, leveraging both features in a collaborative filtering manner;
- METAGL (Park et al., 2022): A representative model selection approach using meta-learning. It uses input multi-modal features as meta-features, where a multi-relational bipartite graph is utilized to estimate the relationship between the input and the choice of models on the task graph;
- NCF++ and METAGL++: In comparison to METAGL and NCF, $M^3$ utilizes additional computation graph descriptions[2] to capture subtask dependencies. To ensure a fair comparison, we extend METAGL and NCF to METAGL++ and NCF++, where original multi-modal input features are enhanced by adding extra text features derived from the computation graph descriptions.

Notably, HuggingGPT (Shen et al., 2023) employs the "In-context Task-model Assignment" model selection strategy. However, our Appendix H experiments show its ineffectiveness, so it is not included in the baselines for simplicity.

**Evaluation metric.** An ideal model selection method for multi-modal reasoning scenarios should allocate the optimal model choice per subtask or subtask type, so as to maximize the execution chance for every input sample. To assess methods on $N_{\text{test}}$ samples from the MS-GQA benchmark, we define the metric Successful Execution Rate (SER) as $\frac{1}{N_{\text{test}}} \sum_{i=1}^{N_{\text{test}}} 1_{\alpha_i}$. Here, $1_{\alpha_i}$ indicates the execution result of the $i$-th sample upon the selected choice of models $\mathbf{c}_i^\star$, either 1 (success) or 0 (fail). SER measures model selector performance: higher SER means better performance and greater overall reliability in multi-modal reasoning. In special cases where all model selection choices either succeed or fail, evaluating the model selector is pointless, so we exclude these cases from our experiments.

**Implementation details.** To facilitate a fair comparison with $M^3$, we use CCE loss in both NCF and METAGL, including their extensions (NCF++ and METAGL++). The default backbone network of the computation graph learner is GAT (Veličković et al., 2017), which is well-known and capable of handling directed acyclic graphs. When dealing with input features with multi-modal information, we utilize "blip-base-vqa" as the default feature extractor. Additionally, the pure textual encoder "bert-base-uncased" is used for encoding the extra computation graph description information. [3]

The results below are reported over five random seeds. The dataset from MS-GQA is split randomly into training, validation, and test sets, with a $6:2:2$ ratio. Ablation studies on the choices of feature extractors and backbone in computation graph learner are deferred to Appendix D.1, D.2 and D.3.

## 4.3 RESULTS: MODEL SELECTION ACROSS DIVERSE TEST DISTRIBUTIONS

In this section, we examine the performance of $M^3$ and other strong baselines across varied test distributions. The test set is divided into multiple sub-test sets. There are two criteria for division: 1) problem structure; 2) model selection difficulty. It is noteworthy that the training dataset remains consistent throughout these experiments. The superiority of $M^3$ is summarized below.

***Training-based* methods outperform *Training-free* methods.** In Table 1 and Table 2, the *Training-based* methods represented by $M^3$ and METAGL significantly outperform and demonstrate greater robustness than the *Training-free* methods represented by EXMETRIC and GLOBALBEST. Specifically, on the *Full* test set in Table 1, *Training-based* methods achieve approximately $64\%$

---

[1]HuggingGPT filters models based on download count on HuggingFace. As our deployed models are not entirely on HuggingFace, we choose the most recently published and largest-parameter model for each subtask.

[2]The LLM decomposes the original input to generate a textual description of the multi-modal reasoning execution process, which we refer to as the computation graph description.

[3]https://huggingface.co/Salesforce/blip-vqa-base, https://huggingface.co/bert-base-uncased

Table 1: **Performance comparison in test scenarios with diverse structural information.** The testing set is split into five sub-test sets: *Query, Choose, Compare, Verify, and Logical*. Each subset maintains consistent structural information; for instance, all *Compare* samples involve tasks related to comparisons. *Full* refers to the uncategorized test set, which is the complete test dataset. "Improv." indicates the specific numerical improvement of $M^3$ compared to the corresponding method in this column.

| Category | | Metrics | Training-free | | | | Training-based | | | | |
| --- | --- | --- | --- | --- | --- | --- | --- | --- | --- | --- | --- |
| | | | RANDOM | VISPROG | EXMETRIC | GLOBALBEST | NCF | NCF++ | METAGL | METAGL++ | $M^3$ |
| Subset | Query | SER (%) | $45.36_{\pm 1.7}$ | $44.85_{\pm 0.0}$ | $58.64_{\pm 0.0}$ | $51.07_{\pm 0.0}$ | $53.94_{\pm 2.9}$ | $53.63_{\pm 4.3}$ | $57.63_{\pm 2.2}$ | $55.18_{\pm 5.2}$ | $59.53_{\pm 1.0}$ |
| | | Improv. (%) | +14.17 | +14.68 | +0.89 | +8.46 | +5.59 | +5.90 | +1.90 | +4.35 | - |
| | Choose | SER (%) | $68.26_{\pm 4.0}$ | $66.94_{\pm 0.0}$ | $68.60_{\pm 0.0}$ | $69.42_{\pm 0.0}$ | $70.25_{\pm 1.7}$ | $73.39_{\pm 2.5}$ | $73.72_{\pm 3.4}$ | $74.71_{\pm 5.1}$ | $76.53_{\pm 2.8}$ |
| | | Improv. (%) | +8.27 | +8.27 | +7.93 | +7.11 | +6.08 | +3.14 | +2.81 | +1.82 | - |
| | Compare | SER (%) | $71.29_{\pm 3.1}$ | $82.26_{\pm 0.0}$ | $61.29_{\pm 0.0}$ | $77.42_{\pm 0.0}$ | $73.92_{\pm 3.9}$ | $74.52_{\pm 2.1}$ | $75.16_{\pm 4.6}$ | $73.55_{\pm 1.8}$ | $76.45_{\pm 1.4}$ |
| | | Improv. (%) | +5.16 | -5.71 | +15.16 | -0.97 | +2.53 | +1.93 | +1.29 | +2.90 | - |
| | Verify | SER (%) | $62.38_{\pm 2.5}$ | $68.40_{\pm 0.0}$ | $65.80_{\pm 0.0}$ | $71.75_{\pm 0.0}$ | $75.46_{\pm 0.9}$ | $71.67_{\pm 2.6}$ | $70.56_{\pm 3.0}$ | $73.01_{\pm 3.0}$ | $75.09_{\pm 0.6}$ |
| | | Improv. (%) | +12.71 | +6.69 | +9.29 | +3.34 | -0.37 | +3.42 | +4.53 | +2.08 | - |
| | Logical | SER (%) | $64.72_{\pm 1.6}$ | $72.47_{\pm 0.0}$ | $59.55_{\pm 0.0}$ | $78.65_{\pm 0.0}$ | $76.50_{\pm 1.8}$ | $77.08_{\pm 1.8}$ | $74.94_{\pm 1.6}$ | $75.73_{\pm 1.7}$ | $77.53_{\pm 2.9}$ |
| | | Improv. (%) | +12.81 | +5.06 | +17.98 | -1.12 | +1.03 | +0.45 | +2.59 | +1.80 | - |
| | Full | SER (%) | $56.51_{\pm 0.3}$ | $59.04_{\pm 0.0}$ | $61.66_{\pm 0.0}$ | $63.58_{\pm 0.0}$ | $65.92_{\pm 1.5}$ | $64.40_{\pm 1.7}$ | $66.01_{\pm 0.4}$ | $65.62_{\pm 3.2}$ | $68.70_{\pm 0.6}$ |
| | | Improv. (%) | +12.19 | +9.66 | +7.04 | +4.12 | +2.83 | +4.30 | +2.69 | +3.08 | - |

to 69% SER, while *Training-free* methods only reach 57% to 64%. Furthermore, Table 1 and Table 2 reveal that in sub-test sets, unlike *Training-free* methods that occasionally excel in one subset while performing poorly in others, *Training-based* methods consistently exhibit overall stability.

$M^3$ **demonstrates its robustness in diverse test sets.** In direct comparisons, Table 1 presents that $M^3$ stands out as the top performer, showcasing a remarkable 2.69% improvement over the previous state-of-the-art (METAGL) in the complete test set (*Full*). Moreover, $M^3$ consistently excels in various sub-test sets across both Table 1 and Table 2, demonstrating its robustness

Table 2: **Performance comparison with respect to the difficulty of model selection.** Each increase in the "difficulty level" indicates a one-unit rise in average model selection difficulty for the sub-test set, leading to a 20% reduction in the executable ratio ($\mathbb{E}\left[\sum_j p_i^j / |c_i|\right]$). Best performances are noted in blue, while the poorest are in orange.

| Difficulty Level | ▪◻◻◻◻ | ▪▪◻◻◻ | ▪▪▪◻◻ | ▪▪▪▪◻ | ▪▪▪▪▪ |
| --- | --- | --- | --- | --- | --- |
| EXMETRIC | $\mathbf{98.35}_{\pm 1.9}$ | $86.67_{\pm 2.9}$ | $59.90_{\pm 3.8}$ | $25.53_{\pm 2.4}$ | $15.62_{\pm 2.8}$ |
| GLOBALBEST | $92.86_{\pm 0.0}$ | $75.46_{\pm 0.0}$ | $65.62_{\pm 0.0}$ | $\mathbf{54.47}_{\pm 0.0}$ | $14.06_{\pm 0.0}$ |
| NCF | $93.12_{\pm 1.9}$ | $82.67_{\pm 2.9}$ | $65.94_{\pm 3.8}$ | $48.08_{\pm 2.4}$ | $15.31_{\pm 2.8}$ |
| METAGL | $82.42_{\pm 3.8}$ | $77.58_{\pm 2.9}$ | $70.10_{\pm 0.9}$ | $53.36_{\pm 3.3}$ | $20.21_{\pm 2.7}$ |
| $M^3$ | $93.30_{\pm 1.7}$ | $84.97_{\pm 1.6}$ | $\mathbf{70.50}_{\pm 2.2}$ | $52.26_{\pm 2.6}$ | $\mathbf{20.42}_{\pm 1.4}$ |

and competitiveness. Even in sub-test sets where it does not claim the top spot, $M^3$ maintains a strong presence, often ranking among the top two or three methods. However, it can be observed that other methods, particularly *training-free* ones, often perform poorly on specific sub-test sets.

$M^3$ **effectively leverages the subtask dependency information.** In Table 1, we assess NCF++ and METAGL++, both of which employ a text encoder to extract textual features describing multi-modal reasoning logic on the task graph. Our experiments reveal that this approach often fails to capture meaningful information and can even harm the performance of the original methods. Conversely, our framework $M^3$, which integrates subtask dependencies into the modeling process as a whole, is more effective and non-trivial.

## 4.4 RESULTS: MODEL SELECTION IN DATA MISSING SCENARIOS

Given the challenges of collecting complete execution results for each sample under all choices of models in real-world scenarios, this section will discuss how *training-based* methods perform on the fixed complete test set (*Full*) in two different types of training data missing scenarios:

- **missing choices of models on the task graph (c)**, which involves varying levels of incomplete execution results associated with different model selection choices per sample. E.g., with a missing ratio of 0.2, $\sim 20\%$ of model selection choices do not have corresponding execution results.
- **missing samples (x)**, where all collected samples have execution results for all model choices on the task graph, but some samples are absent compared to the complete training set. In this case, a missing ratio of 0.2 means that 20% of samples are entirely absent.

**Data missing results in adverse effects.** Figure 5(a) and 5(b) reveal that the missing of data generally leads to performance decline across all *training-based* methods. Specifically, when the missing ratio reaches 0.8, the SER of $M^3$ drops to 66.66% and 64.67%, respectively, while other baselines exhibit a typical 2% to 3% performance decline. Anomalies in some baselines, where performance improves with increased missing ratios, may be attributed to two factors: 1) certain methods are insensitive to training data quantity, and 2) the smaller dataset introduces more experimental randomness.

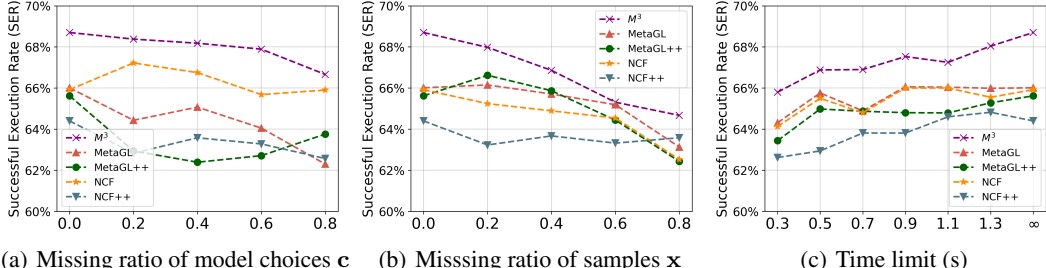

(a) Missing ratio of model choices **c**    (b) Misssing ratio of samples **x**    (c) Time limit (s)

Figure 5: **Performance comparison in scenarios with missing training data and varying time constraints at test-time.** (a) and (b) depict two data-missing scenarios with progressively increasing proportions on the x-axis. (c) illustrates method performance across different time constraints.

**Data missing does not impact the superiority of $M^3$ over baselines.** Figure 5(a) and 5(b) demonstrate that $M^3$ exhibits superior performance compared to other *training-based* baselines in two types of missing scenarios. Specifically, despite the overall decline in performance due to missing data in most methods, $M^3$ consistently outperforms other baselines, underscoring its robustness. Notably, in Figure 5(b), even when up to 80% of samples are missing (64.67%), $M^3$ performs better than the best *training-free* method (GLOBALBEST, 63.58%).

### 4.5    RESULTS: TEST-TIME EFFICIENCY

In practical applications of multi-modal reasoning, beyond enhancing system robustness through model selection, it is crucial to consider the associated cost of implementing the selection process. In this section, we discuss the efficiency of $M^3$, in terms of 1) its extra model selection time and 2) its comparison with baselines under the same inference time limit budget.

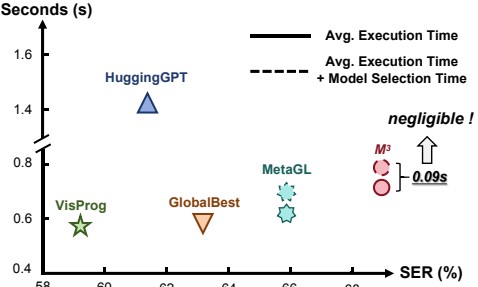

**Negligible runtime overhead at test-time.** Figure 6 illustrates that $M^3$ outperforms all other baselines, particularly *training-free* methods, despite incurring extra time overhead from model selection. However, this additional time overhead (0.09s) is considerably shorter than the overall task execution time (0.62s) and can be considered negligible.

Figure 6: **Comparing $M^3$ and other methods from the perspectives of performance and average time cost.** Execution time encompasses the overall task completion time when agents collaborate using multiple models, while model selection time is the time spent utilizing a proxy for model selection.

$M^3$ **continues to perform the best under various time constraints.** In practical production settings, due to cost constraints, not all models are available during test-time. To mimic this challenge, we gradually decrease the time limit and exclude models from the candidate pool on the task graph if their average execution time exceeds the current limit. Figure 5(c) illustrates the robustness of $M^3$ across a range of scenarios, from the most stringent time limit scenario (0.3s) to scenarios with no time constraints ($\infty$). Despite a decrease in overall performance, $M^3$ consistently excels other baselines, highlighting its suitability for time-constrained settings at test-time.

## 5    CONCLUSION

We introduce $M^3$, a novel framework for assisting autonomous agents in model selection for multi-modal multi-step reasoning scenarios. It tackles the issue of subtask dependencies, a new challenge arising from multi-step reasoning, which existing methods fail to address. In MS-GQA experiments, our framework substantially improves performance, enhancing multi-modal reasoning robustness. Despite resource constraints limiting our experiments to MS-GQA, $M^3$ has broader applications, including subtask node model selection in multi-step reasoning and adaptation to various agents and real-world datasets. For an in-depth discussion of our work's research significance, the advantages and limitations of our method, and an analysis of our experimental results, please refer to Appendix E, F, and G.

ACKNOWLEDGEMENT

We thank anonymous reviewers for their constructive and helpful reviews. This work was supported in part by the National Science and Technology Major Project (No. 2022ZD0115101), the Research Center for Industries of the Future (RCIF) at Westlake University, and the Westlake Education Foundation.

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

CONTENTS OF APPENDIX

## A  DETAILS OF MS-GQA

### A.1  COLLECTION PROCESS AND STATISTICS

MS-GQA is constructed by deploying the VisProg agent (Gupta & Kembhavi, 2023) on the GQA (Hudson & Manning, 2019) dataset, focusing on 9 subtask types, particularly "LOC"(Localization, Text-guided Object Detection) and "VQA" (Visual Question Answering). The other 7 subtask types ("EVAL", "COUNT", "CROP", "CROPLEFT", "CROPRIGHT", "CROPABOVE", "CROPBELOW") involve Python modules, thus excluding the need for model selection. The construction process involves these key steps:

- **Model Zoo Expansion:** VisProg has integrated recently released, popular, open-source models from Visual Question Answering and Text-guided Object Detection domains into its "VQA"and "LOC"modules, expanding the available pool of candidate models.
- **Execution Results Collection:** 10,000 samples are randomly selected from the GQA dataset (test-dev set). Each sample is first decomposed using LLM, resulting in a computation graph description representing the reasoning logic (see "program" in Figure 8). Following this description, all models ("VQA": 7, "LOC": 10, total: $7 \times 10 = 70$) are sequentially executed for each sample, yielding 700,000 execution records. Each execution record represents the result (0: fail, 1: success) of a specific sample's execution for a model selection choice.
- **Data Cleaning**: Samples that experienced reasoning execution failures due to incorrect computation graph descriptions generated by LLM are filtered out. These cases are not within the scope of our model selection research.

Finally, considering resource constraints, we manage to collect a total of 8,426 valid samples, which constitute the foundation for constructing the comprehensive MS-GQA benchmark presented in this study. Within the VisProg framework, the LLM employed is the GPT-3.5-turbo by OpenAI. Regarding the selection of candidate models, we opt for the most recent, widely recognized, high-performing models in each subtask category. The specific candidate models are detailed in Table 3.

Table 3: List of models that can be selected for each subtask in MS-GQA.

| Subtask | Candidate Models | Link | Venue |
|---|---|---|---|
| LOC | owlvit-large-patch14 (Minderer et al.) | https://huggingface.co/google/owlvit-large-patch14 | ECCV 2022 |
| | owlvit-base-patch16 (Minderer et al.) | https://huggingface.co/google/owlvit-base-patch16 | ECCV 2022 |
| | owlvit-base-patch32 (Minderer et al.) | https://huggingface.co/google/owlvit-base-patch32 | ECCV 2022 |
| | glip_large (Zhang et al., 2022a) | https://github.com/microsoft/GLIP | NeurIPS 2022 |
| | glip_tiny_a (Zhang et al., 2022a) | https://github.com/microsoft/GLIP | NeurIPS 2022 |
| | glip_tiny_b (Zhang et al., 2022a) | https://github.com/microsoft/GLIP | NeurIPS 2022 |
| | glip_tiny_c (Zhang et al., 2022a) | https://github.com/microsoft/GLIP | NeurIPS 2022 |
| | glip_tiny_ori (Zhang et al., 2022a) | https://github.com/microsoft/GLIP | NeurIPS 2022 |
| | groundingdino_swinb (Liu et al., 2023b) | https://github.com/IDEA-Research/GroundingDINO | Arxiv 2023 |
| | groundingdino_swint (Liu et al., 2023b) | https://github.com/IDEA-Research/GroundingDINO | Arxiv 2023 |
| VQA | vilt-b32-finetuned-vqa (Kim et al., 2021) | https://huggingface.co/dandelin/vilt-b32-finetuned-vqa | ICML 2021 |
| | git-base-textvqa (Wang et al., 2022) | https://huggingface.co/microsoft/git-base-textvqa | TMLR 2022 |
| | blip-vqa-base (Li et al., 2022) | https://huggingface.co/Salesforce/blip-vqa-base | ICML 2022 |
| | blip2-opt-2.7b (Li et al., 2023) | https://huggingface.co/Salesforce/blip2-opt-2.7b | ICML 2023 |
| | blip2-flan-t5-xl (Li et al., 2023) | https://huggingface.co/sheraz179/blip2-flan-t5-xl | ICML 2023 |
| | instructblip-vicuna-7b (Dai et al., 2023) | https://huggingface.co/Salesforce/instructblip-vicuna-7b | Arxiv 2023 |
| | instructblip-flan-t5-xl (Dai et al., 2023) | https://huggingface.co/Salesforce/instructblip-flan-t5-xl | Arxiv 2023 |

## A.2 FILES OF MS-GQA

There are three main files for MS-GQA:

- **gqa_model_selection_instance_results.json** provides the results of whether a sample can be successfully executed under different model combination choices and the cost time. As shown in Figure 7, the sample with index 1 chooses "instructblip-vicuna-7b" as the "VQA" model and "groundingdino_swint" as the "LOC" model, then this sample can be executed successfully and the cost time is 0.887s.
- **gqa_computation_graph_descrption.json** offers the image ID, problem, and programing text of a sample, as shown in Figure 8.
- **testdev_balanced_questions.json** includes the task type of a sample. As shown in Figure 9, we can get a sample's task type from the attribute of [types][structural].

## A.3 SUBTASK TYPES OF MS-GQA

Table 4 shows the examples of five subtasks, including *Qurey*, *Choose*, *Compare*, *Verify* and *Logical*.

```
{
    "1": [
            {
                "vqa": "instructblip-vicuna-7b",
                "loc": "groundingdino_swint",
                "time": 0.8868060111999512,
                "res": 1
            },
            ...
            {
                "vqa": "blip-vqa-base",
                "loc": "glip_large",
                "time": 0.06340360641479492,
                "res": 1
            }
    ]
}
```

Figure 7: Examples in gqa_model_selection_instance_results.json.

```
{
    "index": 7967,
    "imageId": "n259002",
    "question": "What's underneath the ball?",
    "program":
            "BOX0=LOC(image=IMAGE,object='ball')
            \nIMAGE0=CROP_BELOW(image=IMAGE,box=BOX0)
            \nANSWER0=VQA(image=IMAGE0,question='What is underneath the ball?')
            \nFINAL_RESULT=RESULT(var=ANSWER0)"
}
```

Figure 8: Examples in gqa_computation_graph_descrption.json.

```
{
    "201307251": {
        "semantic": [
            {
                "operation": "select",
                "dependencies": [],
                "argument": "scene"
            },
            {
                "operation": "verify weather",
                "dependencies": [0],
                "argument": "overcast"
            }
        ],
        "entailed": [],
        "equivalent": ["201307251"],
        "question": "Is it overcast?",
        "imageId": "n161313",
        "isBalanced": true,
        "groups": {"global": null, "local": "01-weather_overcast"}, "answer": "no",
        "semanticStr": "select: scene->verify weather: overcast [0]",
        "annotations": {"answer": {}, "question": {}, "fullAnswer": {}},
        "types": {
            "detailed": "weatherVerifyC",
            "semantic": "global",
            "structural": "verify"
        },
        "fullAnswer": "No, it is clear."
    }
}
```

Figure 9: Examples in testdev_balanced_questions.json.

## B  DETAILS OF LOSS CHOICE

In Section 3.2 discussion, we observe that in multi-modal reasoning, a single sample's execution result (success or failure) is known, creating a binary outcome. Notably, the count of successful executions is variable. Consequently, we transform the model selection process in multi-modal reasoning into a

Table 4: Subtask examples in MS-GQA.

| Subtask Type | Question | Image |
|---|---|---|
| *Qurey* | How tall is the chair in the bottom of the photo? |  |
| *Choose* | Is the ground blue or brown? |  |
| *Compare* | Are both the phone and the coffee cup the same color? |  |
| *Verify* | Is the surfer that looks wet wearing a wetsuit? |  |
| *Logical* | Does the utensil on top of the table look clean and black? |  |

multi-label classification issue. Here, the count of positive labels (successful executions) varies, and the total categories are denoted as $|\mathcal{C}|$.

Previous literature often employs Binary Cross-Entropy (BCE) loss for multi-label classification problems due to its ease of optimization. However, BCE's instance-wise nature overlooks the distinct impact of different model selection choices on the same sample.

Therefore, an optimal optimization objective for multi-modal reasoning should emphasize differentiating between model selection choices corresponding to positive and negative labels for a specific sample. Any choice associated with a positive label is deemed optimal. Hence, we choose Categorical Cross-Entropy (CCE) loss since it ensures that "the score for each target class is not lower than the score for each non-target class".

## C  TRAINING DETAILS

NCF, NCF++, and $M^3$ are all implemented by ourselves within a unified pipeline. We employ grid search to filter some crucial hyperparameters. Specifically, we explored hidden sizes [16, 32, 64, 128], learning rates [1e-2, 5e-3, 1e-3, 5e-3, 1e-4], weight decays [0.01, 0.001, 0.0001], and optimizer

options [AdamW, Adam, SGD]. A batch size of 64 is utilized, along with StepLR Scheduler with parameters step size 100 and gamma 0.7.

What's more, we conduct experiments on both METAGL and METAGL++ after adapting the code of MetaGL (Park et al., 2022) to suit our specific scenario. And we train both METAGL and METAGL++ using the optimizer of Adam. The learning rate is adjusted within [1e-2, 5e-3, 1e-3, 5e-3, 1e-4], with a majority of the experiments using 1e-3. The weight decay is set to 0, and the batch size is set to 128.

# D  ABLATION STUDY

## D.1  MULTI-MODAL FEATURE EXTRACTOR

To verify the effect of the quality of multi-modal features on model selection, we perform ablation experiments on the choices of multi-modal feature extractors. We choose BLiP (Li et al., 2022), BERT (Devlin et al., 2018) + ViT (Dosovitskiy et al., 2020), and ViLT (Kim et al., 2021) as the feature extractors separately. As shown in the Table 5, the quality of the features really affects the performance of model selection methods, but $M^3$ stays superiority no matter which feature extractor we use.

Table 5: Performance comparison of model selection methods with different multi-modal feature extractors.

| Feature extractor | NCF | METAGL | $M^3$ |
|---|---|---|---|
| BLiP | $65.92_{\pm 1.5}$ | $66.01_{\pm 0.4}$ | $68.70_{\pm 0.6}$ |
| ViLT | $64.87_{\pm 0.8}$ | $61.90_{\pm 0.8}$ | $65.91_{\pm 0.7}$ |
| BERT+ViT | $63.14_{\pm 0.8}$ | $62.27_{\pm 1.6}$ | $64.94_{\pm 0.5}$ |

## D.2  COMPUTATION GRAPH DESCRIPTION FEATURE EXTRACTOR

NCF++ and METAGL++ leverage extra text features derived from the computation graph descriptions compared to NCF and METAGL. Here we use "bert-base-uncased[4]" (Devlin et al., 2018), "sentence-bert"[5] (Reimers & Gurevych, 2019), and "roberta-base"[6] (Liu et al., 2019) from HuggingFace, respectively, to extract the computational graph features given in textual form by LLM. Table 6 show the performance of NCF++ and METAGL++ across different textual encoders.

Table 6: Performance comparison with different computation graph feature extractors.

| Extractor | NCF++ | METAGL++ |
|---|---|---|
| bert-base-uncased | $64.40_{\pm 1.7}$ | $65.62_{\pm 3.2}$ |
| sentence-bert | $63.95_{\pm 2.2}$ | $64.00_{\pm 1.8}$ |
| roberta-base | $64.28_{\pm 1.3}$ | $65.90_{\pm 1.1}$ |

## D.3  BACKBONE OF COMPUTATION GRAPH LEARNER

In Table 7, we report SER of $M^3$ when using GAT (Veličković et al., 2017), GRU (Chung et al., 2014) and Transformer (Vaswani et al., 2017) as the backbone, respectively. Since GRU and Transformer cannot be directly applied to model directed acyclic graphs, we made corresponding adjustments; however, their performance still lags behind that of GAT.

## D.4  OBJECTIVE FUNCITON

As shown in Table 8, we report the performance comparison of NCF, METAGL and $M^3$ on Binary Cross-Entropy Loss (BCE) and Categorical Cross-Entropy loss (CCE), respectively.

---

[4]https://huggingface.co/bert-base-uncased

[5]https://huggingface.co/sentence-transformers/bert-base-nli-mean-tokens

[6]https://huggingface.co/roberta-base

Table 7: Performance comparison with different backbone for $M^3$.

| Backbone | SER |
|---|---|
| GAT (GNNs) | $68.70_{\pm 0.6}$ |
| GRU (RNNs) | $68.02_{\pm 0.7}$ |
| Transformer | $65.32_{\pm 0.4}$ |

Table 8: Performance comparison with different loss function.

| | NCF | METAGL | $M^3$ |
|---|---|---|---|
| BCE | $64.49_{\pm 1.4}$ | $66.03_{\pm 1.4}$ | $67.65_{\pm 1.3}$ |
| CCE | $65.92_{\pm 2.8}$ | $66.01_{\pm 0.4}$ | $68.70_{\pm 0.6}$ |

# E    RESEARCH SIGNIFICANCE

## E.1    MODEL SELECTION FOR MULTI-MODAL REASONING

- **Model selection techniques have proven successful in various fields.** Model selection techniques, recognized in tasks like time series prediction and graph learning, aim to match samples with suitable models, enabling task completion without sample labels.

- **Errors could have a chain reaction effect.** Choices in models impact overall robustness, crucial in multi-step reasoning with strong task dependencies, as errors can lead to chain reactions on subsequent executions.

- **Underperformance of current model selection strategies.** Section 4's experimental results reveal poor and lacking robustness in existing multi-modal agent model selectors, with methods from other domains yielding unsatisfactory results due to neglect of subtask dependency.

- **A significant gap from the oracle model selector.** In MS-GQA, the oracle model selector attains 100% success, while a random strategy reaches about 56% SER, and existing multi-modal agents usually achieve around 60% effectiveness. Our $M^3$ framework, addressing subtask dependency, enhances results to about 69%. Nevertheless, a noticeable gap from the oracle's 100% remains, underscoring substantial research potential in this area.

## E.2    RELIABLE MODEL SELECTOR: $M^3$

- **Addressed the limited robustness of current model selectors.** The industry emphasizes agent robustness, however, current multi-modal agent research is not mature, and their use of simplistic model selection damages robustness. $M^3$ addresses these shortcomings as an effective and efficient plugin in the model selection stage to enhance overall system robustness.

- **Reliable performance in various scenarios.** $M^3$ demonstrates reliability in various data missing and restriction scenarios on the MS-GQA dataset. Sections 4.4 and 4.5 highlight its superior performance compared to existing training-based methods, reflecting the potential applicability of $M^3$ in real-world production environments.

- **Highly lightweight and efficient.** As pioneers in this research, we've avoided complex network structures and training techniques. Instead, we've tackled a key challenge, subtask dependency, using a straightforward design—a directed acyclic computation graph. This approach models relationships among multi-modal inputs, subtask dependency, and candidate models. The design leads to low costs for both training and testing, with overall memory usage around 6GB, as noted in Section 4.5. This emphasizes $M^3$'s practical potential for real-world production scenarios.

## E.3    PROMISING FUTURE WORK

- **Empower LLM with model selection capabilities.** Currently, the $M^3$ framework operates orthogonally to LLM in multi-modal agents. The former, following task planning by the

latter, performs model selection for each subtask based on its output. Given LLM's robust reasoning abilities, granting it model selection capabilities is a promising endeavor. This would allow LLM to simultaneously handle task planning and model selection, eliminating the need for training an additional model selector and reducing deployment costs in practical production environments.

- **Enhance supervisory signals using intermediate results.** In the current training of the $M^3$ framework, only the final execution results of the multi-modal agent are utilized as supervisory signals, indicating the success of execution or the correctness of the provided answer. However, at each step of reasoning, intermediate results are generated. If these results are judiciously employed as supplementary supervisory signals, we believe it can further improve the effectiveness of $M^3$.

- **More economical unsupervised and semi-supervised training methods.** Similar to the second point, if intermediate results are reasonably utilized, even as a form of data augmentation, supervisory signals can come not only from the final execution results but also from the intermediate stages. This enables a more cost-effective training approach.

### E.4    REAL-WORLD APPLICATIONS

- Utilizing agents to decompose and gradually solve complex multi-modal reasoning tasks is currently one of the mainstream research paradigms for addressing multi-modal challenges. Relevant work, from pioneers like VisProg (Gupta & Kembhavi, 2023) to the highly regarded HuggingGPT (Shen et al., 2023), and more recently LLaVA-Plus (Liu et al., 2023a), has been a focal point for researchers in this field.

- Moreover, AssistGPT (Gao et al., 2023a) and Chameleon (Lu et al., 2023) highlight the potential applications in areas such as video understanding, education, and finance. Meanwhile, inspired by these endeavors (Yang et al., 2023b; Dalal et al., 2023; Wen et al., 2023), we reasonably expect that multi-modal agents whose per-step execution relies on other tools will eventually extend their applications to other AI domains, including autonomous driving, robotics, and embodied intelligence.

- When agents call upon different multi-modal AI models to tackle various subtasks in reasoning, it gives rise to the need for model selection techniques. Specifically, considering

  - the richness and abundance of existing multi-modal model types;
  - the extensive candidate models;
  - the reliability and feasibility demonstrated by model selection in other domains;
  - the overly simplistic or less effective model selection strategies employed by current multi-modal agents;

  Researching model selection in the context of multi-modal reasoning is highly promising and practically valuable in this new scenario.

### E.5    TECHNIQUES

Our technical contributions encompass addressing the challenge of subtask dependency in multi-modal agents by decomposing the original multi-modal task into sub-tasks, as defined in the multi-modal reasoning scenario (see Definition 3.2). Sections 2 and 4 reveal the inadequacy of existing model selection strategies in multi-modal agents, particularly in handling subtask dependencies. Baseline methods, as outlined in Appendix E.1, exhibit notable underperformance compared to the oracle model selector. In response, our proposed model selection framework, $M^3$, skillfully integrates multi-modal inputs, model embeddings, and subtask dependencies on a directed acyclic graph. This approach, detailed from Section 4.3 to Section 4.5, demonstrates the reliability and robustness of $M^3$. As an initial endeavor, the unified modeling approach of $M^3$ holds promise to inspire future researchers.

## F    STRENGTHS AND LIMITATIONS OF THE $M^3$ FRAMEWORK

### F.1    STRENGTHS:

- **Effective performance.** The experimental results in Tables 1 and **??** demonstrate that our approach consistently outperforms baselines across various test distributions on the

MS-GQA dataset, particularly when compared to simplistic strategies like training-free methods.

- **Efficient design.** As pioneers in this domain, we opted for a straightforward yet effective approach. Rather than intricately designing the network structure and optimization strategies for $M^3$, we focused on exploring the critical aspect of subtask dependency. This simplicity is reflected in the efficiency of the entire framework, as evidenced by results in Section 4.5.

- **Applicability across diverse multi-modal agents.** The $M^3$ framework comprises three main components: multi-modal inputs, subtask dependency, and candidate models. Although our experiments were conducted with the VisProg agent, these three inputs constitute foundational components of existing multi-modal agents. For example, in HuggingGPT, if a user inputs an image and a corresponding question, this forms the model's multi-modal inputs. Then, HuggingGPT breaks down the user's original input, identifying various sub-AI tasks and their dependencies, abstracted as subtask dependency. Candidate models in this context refer to all off-the-shelf models or model APIs within HuggingGPT.

### F.2 LIMITATIONS:

- **Data dependency.** The experiments in Section 4.4 demonstrate that, despite $M^3$ performing better than baselines in scenarios with varying degrees of data missing, there is an absolute decline in performance. This underscores the crucial role of data in $M^3$'s effectiveness.

- **Supervisory signal underutilization.** The source of supervisory signals during training is too singular, failing to fully harness the intermediate results of multi-step reasoning. This limitation may contribute to $M^3$'s higher dependency on data.

## G FURTHER ANALYSIS

- $M^3$ **exhibits the best performance on the complete test set of MS-GQA.** Table 1 illustrates that $M^3$ stands out as the top performer overall compared to other baselines. It achieves a significant 2.69% improvement over the previous state-of-the-art (METAGL) in the complete test set (*Full*).

- $M^3$ **continues to perform exceptionally well on the sub-test sets of MS-GQA.** Moreover, in both Table 1 and Table 2, $M^3$ consistently excels in several sub-test sets, including the *Query* and *Choose* sets in Table 1 and the Group 3, and 5 (difficulty level) sets in Table 2. Even in the remaining sub-test sets where it does not perform best, $M^3$ maintains a strong performance, usually ranking among the top two or three methods. In contrast, some baselines may excel in one sub-test set but perform poorly in others. For example, VisProg achieves an 82.26% SER on the Compare sub-test set in Table 1 but fares the worst among all methods on the *Query* (45.36%) and *Choose* (68.26%) sub-test sets. In contrast, the $M^3$ framework consistently performs well across all sub-test sets, which is why $M^3$ significantly outperforms training-free methods on the complete test set (*Full*). This also demonstrates the robustness of $M^3$.

- $M^3$ **depends on the size of dataset.** As shown in Figure 5(b), despite $M^3$ performing better than baselines in scenarios with varying degrees of data missing, there is an absolute decline in performance. Nevertheless, constrained by limited financial resources, we only collected the limited dataset, MS-GQA. So according to the above experimental results, we have reason to believe that expanding the size of the dataset further will improve the performance of $M^3$ on both sub-test sets and the the whole test set.

## H SUPPLEMENTARY EXPERIMENT ON THE "IN-CONTEXT TASK-MODEL ASSIGNMENT" OF HUGGINGGPT

In the original HuggingGPT (Shen et al., 2023) text, there is a mention of utilizing the in-context learning ability of LLM for model selection. Following the configuration in the HuggingGPT source code, we represented each model by using its metadata and relevant descriptions. Based on Table 9, our primary conclusions are as follows:

- The selected model consistently remained the same despite changes in question descriptions or structures.

- The consistency indicates that leveraging the in-context learning capability of LLM currently falls short of achieving genuine and effective dynamic model selection.

Table 9: Distribution of the selected model based on in-context learning

| Candidate Models | Query | Choose | Compare | Logical | Verify |
|---|---|---|---|---|---|
| vilt-b32-finetuned-vqa | 100% | 100% | 100% | 100% | 100% |
| git-base-textvqa | 0% | 0% | 0% | 0% | 0% |
| blip-vqa-base | 0% | 0% | 0% | 0% | 0% |
| blip2-opt-2.7b | 0% | 0% | 0% | 0% | 0% |
| blip2-flan-t5-xl | 0% | 0% | 0% | 0% | 0% |
| instructblip-vicuna-7b | 0% | 0% | 0% | 0% | 0% |
| instructblip-flan-t5-xl | 0% | 0% | 0% | 0% | 0% |

The corresponding experimental details and observations are outlined below:

- **Settings**. We randomly selected 100 questions from GQA, and constructed prompts following HuggingGPT's description. 100 questions cover 5 different question structures, and each question structure typically has distinct reasoning characteristics. The corresponding experimental code is included in the supplementary materials. We randomly selected 100 questions from GQA, and constructed prompts following HuggingGPT's description. 100 questions cover 5 different question structures, and each question structure typically has distinct reasoning characteristics. The corresponding experimental code is included in the supplementary materials.
- **Observations**. All questions, irrespective of task type and question itself, are assigned to the first model (vilt-b32-finetuned-vqa) with five 100% values in the first model column.
- **Comment**. The experiment was conducted only in a one-step scenario, where one model is selected for a single task. Though simple, we believe this case is sufficient to state the fact that "in-context task-model assignment" may not be particularly effective in a multi-step setting.

