# OpenReview forum: "Towards Robust Multi-Modal Reasoning via Model Selection"
_ICLR.cc/2024/Conference — ICLR 2024 poster_

### Official Review · Reviewer_v23h · 2023-10-30

**Soundness:** 3 good
**Presentation:** 2 fair
**Contribution:** 3 good
**Rating:** 6
**Confidence:** 4

**Summary:**

This paper studied the model selection problem in multi-modal reasoning. It first formulated this problem, and then proposed $M^3$ framework as an initial attempt in this field. This paper also created a new dataset, MS-GQA, as a benchmark to compare different methods.

**Strengths:**

1. This paper formulated the model selection problem in multi-modal reasoning, which is a new direction worth investigating.

2. This paper made an initial yet comprehensive effort to study this problem, including a model-selection framework, MS-GQA datasets, as well as a comparison between possible baselines.

**Weaknesses:**

1. The significance of the problem is not well illustrated. While the paper has shown the existence of model selection problem, I am not aware of how important this problem is. There can be lots of problems in multi-modal reasoning, but some may not be of much value. Specifically, is there a huge gap between an oracle model selection and a random selection? Is there a naïve solution that can approach the oracle performance? The authors are suggested to add these preliminary experiments to illustrate the significance of the problem.

2. Lack of ablation study on model inputs. The paper claims that other model selection methods do not take subtask dependency into account. However, the ablation study does not show the effect of using subtask dependency as input. More broadly, because the framework uses various inputs, including multi-modal inputs, node embedding, subtask dependency, a more extensive ablation can be done by removing each component successively. This will show the importance of each component.

**Questions:**

1. Presentation should be improved. I find some of the notations are used without introducing first, which hinders a smooth reading.
Especially in Section 3.2:
* Page 4, bottom line, $\phi \circ \psi$ is used but they have not been introduced.
* Page 5, Section 3.2.1, Para 1, $\psi := [\psi_1, \psi_2]$ is used without explanation.

2. Why choose SER as the metric rather than accuracy? From Figure 1, I thought a wrong model selection will mainly cause the system to give false answer. But the main metric adopted is to measure the successful execution rate. I think there is a difference between successful execution and final accuracy.

3. What is the difficulty level in tab2? How do the authors define the difficulty level?

---

> ### Author Response · Authors · 2023-11-18
> **Response to Reviewer v23h (1/2)**
>
> Dear reviewer v23h:
>
> Thank you very much for the review and feedback. We kindly address your questions as follows:
>
> > **W1:** *The significance of the problem is not well illustrated. While the paper has shown the existence of model selection problem, I am not aware of how important this problem is. There can be lots of problems in multi-modal reasoning, but some may not be of much value. Specifically, is there a huge gap between an oracle model selection and a random selection? Is there a naïve solution that can approach the oracle performance? The authors are suggested to add these preliminary experiments to illustrate the significance of the problem.*
> >
>
> The significance of the problem we introduced to the community and studied in this manuscript can be identified by the performance gap between oracle and our current SOTA method, as illustrated in Section 4.2. In detail,
>
> 1. In our constructed MS-GQA, every sample has at least one model selection choice that would output the correct answer. Therefore, the theoretical upper bound for the entire test set is 100% SER (Successful Execution Rate, where a successful execution in our context indicates that the agent could output the correct answer by executing the selected models);
> 2. As shown in Table 1, a model selector employing random model selection can only achieve approximately 56% SER, while a rigid model selector like VisProg also achieves only 59%, which is not very satisfactory;
> 3. Our method (M3) achieves a SER of 68.7%, significantly surpassing other baselines but still maintaining a gap from the oracle model selector.
>
> In summary, a good, robust model selector can significantly improve SER compared to simple strategies (~56% to 68.7%), but there is still considerable room for improvement compared to the Oracle model selector. This indicates the value of the current research direction.
>
> ---
>
> > **W2:** *Lack of ablation study on model inputs. The paper claims that other model selection methods do not take subtask dependency into account. However, the ablation study does not show the effect of using subtask dependency as input. More broadly, because the framework uses various inputs, including multi-modal inputs, node embedding, subtask dependency, a more extensive ablation can be done by removing each component successively. This will show the importance of each component.*
> >
>
> As discussed in Section 3.2, M3 unifies the key components that emerged in the unexamined scenario of multi-step multi-modal reasoning procedure, namely multi-modal inputs, model (node) embedding, and subtask dependency (see Figure 3).
>
> Below we explain why removing any specific component would make our M3 framework ineffective:
>
> 1. **Multi-modal inputs and model embeddings are irremovable.** Multi-modal inputs and model (node) embeddings are fundamental and basic inputs for model selection, which essentially involves finding the most suitable model for each sample;
> 2. **Subtask dependency is also irremovable.** The representation of subtask dependency through edges in the computation graph is an indispensable component of the computation graph;
>
> In our initial manuscript, we extensively explored the impact of the replaceable components in M3. Specifically, in Appendix D, the ablation study encompasses experiments that span the selection of multi-modal feature extractors to the determination of the objective function.
>
> Besides, we have separately studied the effect of subtask dependency in the conventional model selection methods (namely NCF and MetaGL). It again reveals the uniqueness of our methodology by taking all components as a complete unit.
>
> - Section 4.4 elucidates the advantage of M3 in leveraging subtask dependency information compared to the NCF++ and MetaGL++.

---

> > ### Author Response · Authors · 2023-11-18
> > **Response to Reviewer v23h (2/2)**
> >
> > > **Q1:** *Presentation should be improved. I find some of the notations are used without introducing first, which hinders a smooth reading. Especially in Section 3.2*
> > >
> >
> > Thanks for bringing up the concerns about our notations. We've made necessary adjustments in the latest version of the manuscript (in blue) to address those issues.
> >
> > > **Q2:** *Why choose SER as the metric rather than accuracy?*
> > >
> >
> > We find accuracy to be too general and prefer using SER as a metric, as it attributes each execution result solely to model selection. This allows us to focus on the impact of model selection methods on the overall system.
> >
> > It is important to note that, by default in MS-GQA, cases where model selection leads to execution interruptions are not considered in our manuscript. These samples were removed during preprocessing, and we assume the agent will provide an answer for every model selection choice, which may result in either a successful or unsuccessful execution.
> >
> > > **Q3:** *What is the difficulty level in tab2? How do the authors define the difficulty level?*
> > >
> >
> > In the caption of Table 2, we provided an explanation for the concept of “difficulty level”, which is further detailed as follows:
> >
> > 1. In our experiments in Section 4.3, we categorized all questions into five levels (1, 2, 3, 4, 5).
> > 2. The determination of question difficulty was based on the number of candidate models capable of solving each question.
> > 3. After sorting all questions by difficulty, each level contains the corresponding 20% of samples, with lower level numbers representing questions with the least difficulty among all samples. For example, level 1 corresponds to the 20% of samples with the lowest difficulty in the entire set.

---

> ### Author Response · Authors · 2023-11-22
> **Looking forward to your reply**
>
> Dear Reviewer v23h,
>
> Thank you for your valuable feedback on our manuscript. Upon your request, in our previous response, we have:
>
> - thoroughly discussed the research significance (see Appendix E for more details).
> - provided additional explanations for the settings in the existing ablation study.
> - clarified certain definitions in the original text.
> - rectified instances of non-standard notation usage in the original manuscript.
>
> We would appreciate it if you could let us know if our response has addressed your concerns, and we kindly request a reconsideration of the score.
>
> Best,
>
> Authors

---

### Official Review · Reviewer_1wtR · 2023-11-01

**Soundness:** 3 good
**Presentation:** 3 good
**Contribution:** 2 fair
**Rating:** 5
**Confidence:** 4

**Summary:**

This paper addresses the importance of model selection in multi-modal agents, where Large Language Models (LLMs) play a central role in orchestrating various tools for collaborative multi-step task solving. Unlike traditional methods that use predefined models for subtasks, these multi-modal agents excel by integrating diverse AI models for complex challenges. However, existing multi-modal agents tend to overlook the significance of dynamic model selection, focusing primarily on planning and execution phases, which can make the execution process fragile.

The paper introduces the M3 framework as a plug-in with a small runtime overhead at test time. This framework aims to enhance model selection and improve the robustness of multi-modal agents in multi-step reasoning scenarios. In the absence of suitable benchmarks, the authors create a new dataset called MS-GQA, designed to investigate the model selection challenge in multi-modal agents. The experiments demonstrate that the M3 framework enables dynamic model selection by considering both user inputs and subtask dependencies, ultimately enhancing the overall reasoning process. The authors plan to make their code and benchmark publicly available.

**Strengths:**

The paper provides a clear analysis of the challenges.
Besides the method, the paper also provides a dataset as one of the contributions.
The experimental results show significant improvements.

**Weaknesses:**

The method uses a heuristic process to perform selection which the capacity is relying on the pre-trained models themselves.
How about the generalization capacity for the zero-shot tasks?

**Questions:**

refer to the above content.

---

> ### Author Response · Authors · 2023-11-18
> **Response to Reviewer 1wtR**
>
> Dear reviewer 1wtR:
>
> Thank you very much for the review and feedback. We kindly address your questions as follows:
>
> > **W1.1**: *The method uses a heuristic process to perform selection which the capacity is relying on the pre-trained models themselves.*
> >
>
> We are uncertain about the concept of “**heuristic process”** and “**pre-trained models”**  mentioned by the reviewer. It would be very appreciated if the reviewer could elaborate on it a bit.  A more detailed explanation would greatly assist us in addressing any potential misunderstandings and further refining our work accordingly.
>
> Besides, we would like to clarify that our study isolates the effect of multi-step reasoning quality—-caused by the LLMs—-and focuses on the unexplored model selection scenario for the given reasoning paths. We aim to utilize model selection for each sub-task to further improve the robustness of the multi-modal agent system.
>
> > **W1.2**: *How about the generalization capacity for the zero-shot tasks?*
> >
> As highlighted in [here](https://openreview.net/forum?id=KTf4DGAzus&noteId=wuDiUQJVxC), creating a novel model selection benchmark is a non-trivial task. Because constrained by the resources, we paid our efforts only on collecting an MS-GQA dataset and haven’t tested the generalization capacity for other zero-shot tasks. We acknowledge the importance of assessing generalization across diverse datasets and plan to incorporate widely recognized datasets in our future research to thoroughly investigate M3's performance in handling zero-shot tasks.

---

### Official Review · Reviewer_JtbH · 2023-11-04

**Soundness:** 3 good
**Presentation:** 3 good
**Contribution:** 3 good
**Rating:** 6
**Confidence:** 4

**Summary:**

This paper addresses the need for improved model selection in multi-modal agents to enhance their robustness in multi-step reasoning tasks. The authors introduce the M3 framework to facilitate dynamic model selection, considering user inputs and subtask dependencies, and they present the MS-GQA dataset as a benchmark for evaluating their framework's performance.

**Strengths:**

Identification of Critical Challenge: The paper recognizes and addresses a significant challenge in multi-modal agents, which is the selection of appropriate models for subtasks, a crucial aspect often overlooked in prior research.

Introduction of the M3 Framework: The paper presents the M3 framework, which aims to improve model selection by considering user inputs and subtask dependencies. The framework is designed with negligible runtime overhead at test-time, making it practical for real-world applications.

Creation of the MS-GQA Dataset: The authors introduce the MS-GQA dataset, specifically designed for investigating model selection challenges in multi-modal agents. This dataset is a valuable resource for benchmarking and advancing research in this area.

Experimental Findings: The paper provides experimental evidence that the M3 framework enhances dynamic model selection and, as a result, bolsters the overall robustness of multi-modal agents in multi-step reasoning tasks.

**Weaknesses:**

Limited Baseline Comparison: The paper could benefit from a more comprehensive comparison of the M3 framework with existing methods. While it claims to outperform traditional model selection methods, a detailed comparison with state-of-the-art techniques would provide a more robust evaluation.

Insufficient Experimental Discussion: The discussion of experimental results could be more in-depth. The paper does not thoroughly analyze the scenarios where the M3 framework performs exceptionally well or falls short. A deeper dive into the results would provide valuable insights into the framework's strengths and limitations.

Real-World Application Discussion: While the paper discusses the practicality of the M3 framework, it could delve further into real-world applications or use cases where this framework could be deployed effectively. This would provide a clearer vision of its potential impact.

**Questions:**

Please see weakness.

---

> ### Author Response · Authors · 2023-11-18
> **Response to Reviewer JtbH (1/2)**
>
> Dear reviewer JtbH:
>
> Thank you very much for the review and feedback. We kindly address your questions as follows:
>
> > **W1**: *Limited Baseline Comparison: The paper could benefit from a more comprehensive comparison of the M3 framework with existing methods. While it claims to outperform traditional model selection methods, a detailed comparison with state-of-the-art techniques would provide a more robust evaluation*.
> >
>
> We investigate model selection in multi-step multi-modal reasoning scenarios, a novel research direction that has not been explored previously. Currently, there are no existing methods available for direct use; therefore, all model selection methods in our current baselines are adapted from related domains. We have organized the baselines as follows:
>
> 1. For training-free methods, we include classic approaches like GLOBALBEST and other simplistic model selection strategies employed by existing multi-modal agents.
> 2. For training-based methods, we present two representative model selection baselines—collaborative filtering-based NCF and meta-learning-based MetaGL. Additionally, for a fairer comparison, we extend NCF and MetaGL to NCF++ and MetaGL++, examining the significance of subtask dependency.
>
> It would be appreciated if the reviewer could specify suitable references as baselines, and we will include them upon request.
>
> > **W2**: *Insufficient Experimental Discussion: The discussion of experimental results could be more in-depth. The paper does not thoroughly analyze the scenarios where the M3 framework performs exceptionally well or falls short. A deeper dive into the results would provide valuable insights into the framework's strengths and limitations*.
> >
>
> We have added a more in-depth experimental analysis to reveal the framework's strengths and limitations in our new manuscript. More analysis can be found in Appendix G. We also summarize some key points below.
>
> 1. When does M3 perform exceptionally well:
>
>     **Observations:**
>
>     - M3 exhibits the best performance on the complete test set (Full) of MS-GQA.
>     - M3 continues to perform exceptionally well on the sub-test sets of MS-GQA.
>
>     **Analysis:**
>
>     Compared with other methods, M3 framework's success can be attributed to its unique approach of integrating multi-modal inputs, model (node) embedding, and subtask dependency to model the intricate information flow within the multi-step reasoning process.
>
>
> 2. When does M3 fall short:
>
>     **Observations:**
>
>     - M3 may perform less effectively than some training-free baselines in some sub-test sets of MS-GQA.
>
>     **Analysis:**
>
>     - The training dataset size may affect the performance of training-based methods including M3. As shown in Figure 5 (b), despite M3 performing better than baselines in scenarios with varying degrees of data missing, there is an absolute decline in performance.
>     - Preferences for models vary across different sub-test sets and subtask types. The training-free baseline tends to fixate on a specific model for each subtask type, meaning some training-free methods may perform well on one sub-test set (but may poorly on others).
>
>     We apologize that constrained by the resources, we only collected an MS-GQA dataset with ~8k samples. However, according to Figure 5 (b), we have reason to believe that expanding the size of the dataset further will improve the performance of M3.

---

> ### Author Response · Authors · 2023-11-18
> **Response to Reviewer JtbH (2/2)**
>
> > **W3**: *Real-World Application Discussion: While the paper discusses the practicality of the M3 framework, it could delve further into real-world applications or use cases where this framework could be deployed effectively. This would provide a clearer vision of its potential impact.*
> >
> The practical utility of M3 lies in enhancing the reasoning robustness of multi-modal agents, thereby facilitating their application across various popular domains in the real world. In detail,
>
> - Breaking down and solving complex multi-modal tasks step by step using agents is a leading research approach in addressing multi-modal challenges.
> - [1,2] highlight the prospective applications of multi-modal approaches in areas such as video understanding, finance, and education. Meanwhile, the advancements presented in [3,4,5] have given rise to the potential application of multi-modal agents with per-step tool reliance in domains like robotics and embodied intelligence.
> - When agents call upon different multi-modal AI models to tackle various subtasks in reasoning, it gives rise to the need for model selection techniques. Specifically, considering:
>     1. the richness and abundance of existing multi-modal model types;
>     2. the extensive candidate models;
>     3. the reliability and feasibility demonstrated by model selection in other domains;
>     4. the overly simplistic or less effective model selection strategies employed by current multi-modal agents;
>
>     Researching model selection in the context of multi-modal reasoning is highly promising and practically valuable in this new scenario.
>
> [1] Gao, Difei, et al. "AssistGPT: A General Multi-modal Assistant that can Plan, Execute, Inspect, and Learn." *arXiv preprint arXiv:2306.08640* (2023).
>
> [2] Lu, Pan, et al. "Chameleon: Plug-and-play compositional reasoning with large language models." *arXiv preprint arXiv:2304.09842* (2023).
>
> [3] Yang, Jingkang, et al. "Octopus: Embodied Vision-Language Programmer from Environmental Feedback." *arXiv preprint arXiv:2310.08588* (2023).
>
> [4] Dalal, Murtaza, et al. "Plan-Seq-Learn: Language Model Guided RL for Solving Long Horizon Robotics Tasks." *CoRL 2023 Workshop on Learning Effective Abstractions for Planning (LEAP)*. 2023.
>
> [5] Wen, Licheng, et al. "On the Road with GPT-4V (ision): Early Explorations of Visual-Language Model on Autonomous Driving." *arXiv preprint arXiv:2311.05332* (2023).

---

> ### Author Response · Authors · 2023-11-22
> **Looking forward to your reply**
>
> Dear Reviewer JtbH,
>
> We appreciate your valuable feedback on our manuscript. Upon your request, in our previous response, we have:
>
> - provided explanations for the settings in the existing baselines.
> - expanded our discussions on the experiments.
> - delved deeper into real-world applications and research significance.
>
> We would appreciate it if you could let us know if our response has addressed your concerns, and we kindly request a reconsideration of the score.
>
> Best,
>
> Authors

---

### Official Review · Reviewer_JZBq · 2023-11-07

**Soundness:** 3 good
**Presentation:** 3 good
**Contribution:** 2 fair
**Rating:** 5
**Confidence:** 3

**Summary:**

This paper concentrates on the issue of model selection for multi-modal reasoning tasks. It introduces the Model Selector for the Multi-Modal Reasoning (M3) framework, designed to model the dependencies among subtasks and enhance model selection in multi-modal reasoning. To explore the model selection challenge in multi-modal tasks, the authors have created the MS-GQA dataset. The experiments demonstrate that the M3 framework improves the robustness of reasoning on the MS-GQA dataset.

**Strengths:**

1. This paper adeptly formulates the model selection problem within multi-modal reasoning contexts and constructs the MS-GQA dataset.
2. The paper is well-founded in its pursuit to address the overlooked subtask dependencies in previous works. The proposed M^3 framework innovatively and effectively models the relationship between samples, selected models, and subtask dependencies.
3. The experiments conducted on MS-GQA demonstrate the efficiency and efficacy of the M^3 framework.

**Weaknesses:**

1. The primary concern is that model selection is a small part of multi-modal reasoning. It remains to be seen whether it is important for the entire task and how it can benefit real-world applications. The selection method proposed in this paper involves complex proxy training and may need to be more universally applicable or scalable for different reasoning tasks.

2. Lack of reproducibility: The paper must include crucial details, such as the LLM used. The constructed MS-GQA dataset is not yet open-sourced, and the paper fails to provide even a single example of the dataset. Furthermore, the paper does not demonstrate how the proposed methods can improve various reasoning tasks and whether they can be applied to open-source models like LLaMA.

3. The implementation of the baselines is weak: The original HuggingGPT paper dynamically selected models for tasks through in-context task-model assignment, yet this paper describes it as only using "external metrics" and implements it as "choosing the most recently published one for each subtask", which is misleading and causes unfair comparisons.

4. The experiments could be more convincing: This paper only reports results on a newly created MS-GQA dataset. Even though it's compared to simple baselines that do not require training, like directly using the latest or best models (as shown in Table 2), the proposed M^3 method does not show consistent improvements and may even significantly degrade performance. It would be more convincing if experiments were conducted on more reasoning tasks, as done in VisProg and HuggingGPT.

**Questions:**

See weakness.

---

> ### Author Response · Authors · 2023-11-18
> **Response to Reviewer JZBq (1/3)**
>
> Dear reviewer JZBq:
>
> Thank you very much for the review and feedback, we kindly address your questions as follows.
>
> > **W1.1:** *The primary concern ... can benefit real-world applications.*
> >
> We summarize and elucidate the value of exploring model selection in multi-modal reasoning scenarios from four specific perspectives:
>
> 1. In Section 2.2, we demonstrate the proven value of model selection in other domains (e.g. outlier detection, time-series prediction, and graph learning);
> 2. Table 1 provides experimental evidence, showing that model selection methods successful in other domains do not perform well in multi-modal reasoning scenarios under the agent. Additionally, our analysis in Section 4.3 further indicates that effectively leveraging subtask dependency, unique to multi-step reasoning, significantly improves the successful execution rate of the overall system;
> 3. Existing methods still have a significant gap from the oracle model selector, see Appendix E.1 for more details;
> 4. A newly added Appendix E.4 and our joint reply discusses the necessity for a reliable model selector in current multi-modal agents. This enhances the overall robustness of the agent system, facilitating their application in real-world domains like finance, education, robotics, and embodied intelligence.
>
> > **W1.2:** *The selection method proposed in this paper involves complex proxy training*
> >
> Certainly, proxy training introduces additional overhead, but:
> 1. As indicated in our analysis in Section 4.3, training-free methods exhibit a noticeable performance gap compared to training-based approaches.
> 2. Still in Section 4.3, the SOTA model selection methods in other domains, though being training-based, fall short of addressing the multi-modal reasoning scenarios.
> 3. As an initial attempt in the field, our proposed framework (M3) incorporates all necessary components required to model the dependent reasoning process for model selection. Our simple design takes less than an hour of training. Section 4.5 further clarifies the efficiency of our method during test-time.
> 4. As our next step, we are working on reducing the overhead of proxy training from the aspect of ability assignment and in-context learning. This exploration is beyond the scope of our current manuscript.
>
> > **W2.4:** *whether they can be applied to open-source models like LLaMA*
> >
> > **W2.1:** *The paper must include crucial details, such as the LLM used*
> >
> In a multi-modal agent scenario, the LLM's role is to decompose GQA questions into AI subtasks and determine their execution order. In our case, we utilize the LLM to construct the MS-GQA dataset, and our method design and experimental results are irrelevant to the LLM. Note that,
> - Our study isolates the effect of multi-step reasoning quality—-caused by the LLMs—-and focuses on the unexplored model selection scenario for the given reasoning paths.
> - As explained in our Section 3, M3 relies on the generated reasoning path from a LLM, and performs the model selection along the path per subtask node.
> - We have elaborated the LLM details in the Appendix A (of the original submission), where we use **GPT-3.5 turbo** as our default LLM.
>
> We agree with the reviewer that the significance of model selection would be more pronounced when the quality of the reasoning path cannot be guaranteed, e.g., with open-source models. We leave this part for future work.
>
> > **W2.2:** *The constructed MS-GQA dataset is not yet open-sourced, and the paper fails to provide even a single example of the dataset*
> >
> We have open-sourced our constructed dataset and source code in the supplementary materials.
>
> Statistical information about MS-GQA can be found in Sec 4.1 and Appendix A. Additionally, we have updated the introduction of dataset files and provided examples of samples in Appendix A.2 and A.3. The dataset files are also available in the supplementary materials.
>
> We also commit to promptly releasing the code and dataset as indicated in our abstract.
>
> > **W2.3:** *the paper does not demonstrate how the proposed methods can improve various reasoning tasks*
> >
> To showcase the robustness of our method in handling different reasoning tasks:
> 1. We categorized the existing MS-GQA dataset into five subgroups/tasks (Query, Choose, Compare, Verify, and Logical) based on question structures, each with distinct reasoning characteristics (see Appendix A.3 for examples).
> 2. We also determined the question difficulty by considering the number of candidate models capable of solving each question. Questions solvable by any model selection choices were deemed *the easiest*, while those with limited viable choices were considered *challenging*. Then, questions were categorized by their difficulty levels.
>
> In both scenarios, the results in Tables 1 and 2, along with the analysis in Section 4.3, demonstrate the robust performance of our method across various reasoning tasks, surpassing other baselines.

---

> ### Author Response · Authors · 2023-11-18
> **Response to Reviewer JZBq (2/3)**
>
> > **W1.3:** *may need to be more universally applicable or scalable for different reasoning tasks*
> >
> > **W4.2:** *It would be more convincing if experiments were conducted on more reasoning tasks, as done in VisProg and HuggingGPT.*
> >
> As the very initial exploration of the community, we established a benchmark and introduced MS-GQA, a significant dataset derived from GQA.
>
> - We assert MS-GQA's representativeness for model selection for multi-modal reasoning' given GQA's wide recognition in the Visual Question Answering community [3,4,5] and its use in testing multi-modal agents like VisProg [1] and ViperGPT [2].
> - Additionally, to validate our framework on diverse test data, we separately divided MS-GQA into five sub-datasets with different question structures and five sub-datasets with varying difficulty levels. (see Sec 4.3)
> - Furthermore, creating a novel model selection dataset is a non-trivial task involving several steps:
>     1. finding a publicly accessible multi-modal reasoning dataset covering various reasoning types;
>     2. identifying the dataset's subtask types (e.g., object detection, segmentation, and visual question answering);
>     3. deploying popular candidate models for different subtask types;
>     4. utilizing existing multi-modal agents to execute the dataset with diverse model selection choices;
>     5. post-processing the results, discarding or revising samples with unsuccessful executions due to engineering problems or errors in the LLM-generated task execution graph.
>
>     This entire process is laborious and time-consuming, not to mention the significant costs associated with invoking LLM (GPT-3.5-turbo).
>
>
> While we recognize the limitation of conducting experiments solely on MS-GQA, it is important to note that: 1) MS-GQA serves as a representative dataset, 2) our method consistently performs well across various test scenarios, highlighting its robustness, and 3) developing a new model selection dataset presents substantial challenges. Due to these factors, we cannot provide experimental results for different multi-modal reasoning tasks at the moment.
>
> We are planning to include other widely recognized datasets in future research to strengthen the persuasiveness of our findings, considering our resource limitations.
>
> [1] Gupta, Tanmay, and Aniruddha Kembhavi. "Visual programming: Compositional visual reasoning without training." *Proceedings of the IEEE/CVF Conference on Computer Vision and Pattern Recognition*. 2023.
>
> [2] Surís, Dídac, Sachit Menon, and Carl Vondrick. "Vipergpt: Visual inference via python execution for reasoning." *arXiv preprint arXiv:2303.08128* (2023).
>
> [3] Li, Junnan, et al. "Blip-2: Bootstrapping language-image pre-training with frozen image encoders and large language models." *arXiv preprint arXiv:2301.12597* (2023).
>
> [4] Tiong, Anthony Meng Huat, et al. "Plug-and-play vqa: Zero-shot vqa by conjoining large pretrained models with zero training." *arXiv preprint arXiv:2210.08773* (2022).
>
> [5] Tan, Hao, and Mohit Bansal. "Lxmert: Learning cross-modality encoder representations from transformers." *arXiv preprint arXiv:1908.07490* (2019).
>
> > **W4.1:** *Even though it's compared to simple baselines that do not require training, like directly using the latest or best models (as shown in Table 2), the proposed M^3 method does not show consistent improvements and may even significantly degrade performance.*
> >
>
> Training-free methods may perform well on one sub-test set (If their choice happens to align with the preference) but poorly on others, while the M3 framework consistently performs well across all sub-test sets, as mentioned in our paper Section 4.3:
>
> - First of all, preferences for models vary across different sub-test sets and subtask types.
> - The training-free baseline tends to fixate on a specific model for each subtask type, meaning some training-free methods may perform well on one sub-test set (e.g., VisProg performs best in the Compare sub-test set) but poorly on others (e.g., VisProg performs worst in the Query and Choose sub-test sets).
> - Tables 1 and 2 demonstrate that M3 performs best on many sub-test sets and still exhibits strong performance on the remaining sets, showing its robustness.
>
> The experimental results in Section 4.4 also demonstrate a notable improvement in all training-based methods with an increase in the number of samples. Therefore, although M3 may **occasionally** perform less effectively than training-free methods on the current scale of the MS-GQA dataset, we have reason to believe that M3 will exhibit better performance on larger datasets in the future.

---

> ### Author Response · Authors · 2023-11-18
> **Response to Reviewer JZBq (3/3)**
>
> > **W3:** The implementation of the baselines is weak: The original HuggingGPT paper dynamically selected models for tasks through in-context task-model assignment, yet this paper describes it as only using "external metrics" and implements it as "choosing the most recently published one for each subtask", which is misleading and causes unfair comparisons.
> >
>
> Thank you for highlighting the issue in our HuggingGPT baseline implementation. We agree with the reviewer that the original HuggingGPT uses in-context learning for model assignment, rather than directly selecting models based on “external metrics”. We have made revisions in blue in the new version.
>
> Moreover, we conducted experiments on the "in-context task-model assignment" model selection strategy of HuggingGPT, and displayed the partial results in the table below:
> - The selected model consistently remained the same despite changes in question descriptions or structures.
> - The consistency indicates that leveraging the in-context learning capability of LLM currently falls short of achieving genuine and effective dynamic model selection.
> |  | model 1: vilt-b32-finetuned-vqa | model 2: git-base-textvqa | model 3: blip-vqa-base  | model 4: blip2-opt-2.7b | model 5: blip2-flan-t5-xl | model 6: instructblip-vicuna-7b | model 7: instructblip-flan-t5-xl |
> | --- | --- | --- | --- | --- | --- | --- | --- |
> | Query | 100% | 0% | 0% | 0% | 0% | 0% | 0% |
> | Choose | 100% | 0% | 0% | 0% | 0% | 0% | 0% |
> | Compare | 100% | 0% | 0% | 0% | 0% | 0% | 0% |
> | Logical | 100% | 0% | 0% | 0% | 0% | 0% | 0% |
> | Verify | 100% | 0% | 0% | 0% | 0% | 0% | 0% |
>
> The corresponding experimental details and observations are outlined below:
>
> - Settings. We randomly selected 100 questions from GQA, and constructed prompts following HuggingGPT's description. 100 questions cover 5 different question structures, and each question structure typically has distinct reasoning characteristics. The corresponding experimental code is included in the supplementary materials.
> - Observations. All questions, irrespective of task type and question itself, are assigned to the first model (vilt-b32-finetuned-vqa) with five 100% values in the first model column.
> - Comment. The experiment above was conducted only in a one-step scenario, where one model is selected for a single task. Though simple, we believe this case is sufficient to state the fact that “in-context task-model assignment” may not be particularly effective in a multi-step setting.

---

> ### Author Response · Authors · 2023-11-21
> **Looking forward to your reply**
>
> Dear Reviewer JZBq,
>
> We appreciate your valuable feedback on our manuscript. Upon your request, in our previous response, we have:
>
> - delved deeper into real-world applications and research significance;
> - provided code and dataset examples in the supplementary materials and the revised manuscript to enhance reproducibility;
> - included additional details regarding baseline settings;
> - expanded our discussions on the experiments;
>
> We would appreciate it if you could let us know if our response has addressed your concerns, and we kindly request a reconsideration of the score.
>
> Best,
>
> Authors

---

### Author Response · Authors · 2023-11-18
**Reply to all reviewers (1/2)**

## **Summary of revision**

First of all, we sincerely appreciate the reviewers for their valuable time and constructive comments. We have thoroughly considered their suggestions and diligently incorporated them into our revised manuscript. For the convenience of the reviewers, we outline the key modifications of the manuscript in the following summary:

- [Reviewer **JZBq** and **v23h**] We added a new section Appendix E, here we expanded the discussion on *Research Significance* from four perspectives: 1) model selection for multi-modal reasoning, 2) our proposed M3 framework, 3) promising future work, and 4) real-world applications;
- [Reviewer **JZBq**] We have included our constructed dataset and source code in the supplementary materials to support the reproducibility of our project. Also, We commit to promptly releasing the code and dataset as indicated in our abstract.
- [Reviewer **JZBq**] We clarified that the M3 framework functions independently of LLM in multi-modal agents. Additionally, in Appendix A, we stated that our current LLM is GPT-3.5-turbo.
- [Reviewer **JZBq**] We revised the definition of the HuggingGPT baseline in Sec 4.2 in blue and conducted additional experiments to check the effectiveness of the "in-context task-model assignment” of the original HuggingGPT, detailed in [this link](https://openreview.net/forum?id=KTf4DGAzus&noteId=1HU0jyBLYU).
- [Reviewer **JZBq** and **JtbH**]  We added a new subsection Appendix E.4, to include the discussions about real-world applications of model selection in multi-modal agents.
- [Reviewer **JZBq**, **JtbH**, and **v23h**] We added a new section, Appendix G, as a supplement, to delve deeper into the discussion of experimental results.
- [Reviewer **JtbH**] We responded and clarified the rationale for the current baseline setting [here](https://openreview.net/forum?id=KTf4DGAzus&noteId=AaX4iFrgbF).
- [Reviewer **v23h**] We responded and clarified the rationale for the current ablation study setting [here](https://openreview.net/forum?id=KTf4DGAzus&noteId=LqGkHgYjwM), and mentioned the ablation study we have done in Appendix D of our original manuscript.
- [Reviewer **JtbH**] We added a new section Appendix F, to include the discussions about the strengths and limitations of our proposed method.

## **Significance, novelty, and broader impacts of our manuscript**

We would like to emphasize and clarify the importance of our work, as acknowledged by the positive review from all reviewers:

1. [Reviewer ****v23h****] We pioneered model selection for multi-modal reasoning.
2. [Reviewer ****JZBq****, ****JtbH****, and ****v23h****] We recognized/defined the model selection problem within the context of multi-modal reasoning.
3. [Reviewer ****JZBq****, ****JtbH****, ****1wtR****, and ****v23h****] We introduced a new benchmark, along with a newly created dataset MS-GQA.
4. [Reviewer **JZBq**, **1wtR**, and ****JtbH****] We highlighted subtask dependency (overlooked by previous research) as a critical challenge.
5. [Reviewer **JZBq** and ****JtbH****] We constructed a novel model selection framework, M3, which ingeniously models relationships among samples, selected models, and subtask dependencies.
6. [Reviewer **JZBq** and ****JtbH****] Experiments on MS-GQA validate the efficiency and effectiveness of the M3 framework.

Furthermore, in order for readers to better understand the purpose of our proposed new task, "model selection in multi-modal reasoning", we here re-examine and reclaim the significance and broader impacts of our research from vision, application, and technique perspectives:

**Vision:**

- In the current era of AI, relying solely on a single model is insufficient to fully support real-world applications. Taking LLM as an example:
    1. RAG facilitates LLM in accessing reliable and real-time information, as well as managing private data.
    2. Chain-of-thoughts technique significantly improves the model's reasoning capabilities.
    3. Further, we can empower LLM to engage various models—language, speech, vision—to collectively tackle intricate application tasks.
- In the upcoming AI era, diverse technologies and models will collaborate synergistically, forming a framework akin to an OS (Operating System). This framework will enhance AI application development and model operation, offering developers and users a comprehensive platform for interactions.
- Just as mobile OS lets you interact with ads on video sites, make purchases, and complete transactions, AI OS allows the use of various AI models or apps for distinct sub-tasks. Similar to how video sites, shopping platforms, and payment software naturally create a task chain/graph on the mobile OS to fulfill a "purchase recommended products from a video site" request, different AI models within AI OS will also form a task chain/graph.

---

### Author Response · Authors · 2023-11-18
**Reply to all reviewers (2/2)**

- Similar to selecting video websites, e-commerce platforms, and payment software on mobile OS, AI OS will have a justified need for model selection. Moreover, the interplay of apps on mobile OS aligns with the task dependency concept in AI OS.
- Drawing from this abstraction in AI OS and contrasting it with real-world situations on mobile OS, we anticipate the impactful application potential of model selection technology in the future AI era.

**Application:**

- Utilizing agents to decompose and gradually solve complex multi-modal reasoning tasks is currently one of the mainstream research paradigms for addressing multi-modal challenges. Relevant work, from pioneers like VisProg [1] to the highly regarded HuggingGPT [2], and more recently LLaVA-Plus [3], has been a focal point for researchers in this field.
- Moreover, AssistGPT [7] and Chameleon [8] highlight the potential applications in areas such as video understanding, education, and finance. Meanwhile, inspired by these endeavors [4,5,6], we reasonably expect that multi-modal agents whose per-step execution relies on other tools will eventually extend their applications to other AI domains, including autonomous driving, robotics, and embodied intelligence.
- When agents call upon different multi-modal AI models to tackle various subtasks in reasoning, it gives rise to the need for model selection techniques. Specifically, considering
    1. the richness and abundance of existing multi-modal model types;
    2. the extensive candidate models;
    3. the reliability and feasibility demonstrated by model selection in other domains;
    4. the overly simplistic or less effective model selection strategies employed by current multi-modal agents;

    Researching model selection in the context of multi-modal reasoning is highly promising and practically valuable in this new scenario.


**Technique:**

- In multi-modal agents, decomposing the original multi-modal task into sub-tasks introduces subtask dependency, a key challenge defined in the multi-modal reasoning scenario (see Definition 3.2).
- Sec 2 and 4 indicate that existing model selection strategies in multi-modal agents are usually simplistic and ineffective. Sec 4.3 states that traditional model selection methods struggle to adequately handle the dependency between subtasks in the new scenario (multi-modal reasoning).
- Existing methods (baselines) are currently underperforming below expectations, with a noticeable gap compared to the oracle model selector. Refer to Appendix E.1 for details.
- Our novel model selection framework, M3, adeptly integrates multi-modal inputs, model embeddings, and subtask dependency on a directed acyclic graph. This straightforward and effective modeling approach considers the unique characteristics of multi-modal reasoning, as evidenced in the experiments from Sec 4.3 to Sec 4.5, highlighting the reliability and robustness of M3.
- The unified modeling approach of M3, as an initial endeavor, is poised to inspire future researchers.

[1] Gupta, Tanmay, and Aniruddha Kembhavi. "Visual programming: Compositional visual reasoning without training." *CVPR* (2023).

[2] Shen, Yongliang, et al. "Hugginggpt: Solving ai tasks with chatgpt and its friends in huggingface." *arXiv* (2023).

[3] Liu, Shilong, et al. "LLaVA-Plus: Learning to Use Tools for Creating Multimodal Agents." *arXiv* (2023).

[4] Yang, Jingkang, et al. "Octopus: Embodied Vision-Language Programmer from Environmental Feedback." *arXiv* (2023).

[5] Dalal, Murtaza, et al. "Plan-Seq-Learn: Language Model Guided RL for Solving Long Horizon Robotics Tasks." *CoRL 2023 Workshop on Learning Effective Abstractions for Planning (LEAP)*. 2023.

[6] Wen, Licheng, et al. "On the Road with GPT-4V (ision): Early Explorations of Visual-Language Model on Autonomous Driving." *arXiv* (2023).

[7] Gao, Difei, et al. "AssistGPT: A General Multi-modal Assistant that can Plan, Execute, Inspect, and Learn." *arXiv* (2023).

[8] Lu, Pan, et al. "Chameleon: Plug-and-play compositional reasoning with large language models." *arXiv* (2023).

---

### Meta-Review · Area_Chair_GAaa · 2023-12-17

**Metareview:**

This paper proposes an interesting pipeline that can utilize agents to decompose and gradually solve complex multi-modal reasoning tasks for addressing multi-modal challenges. This research direction is quite inspiring and important to next AI era. The dynamic model selection, considering user inputs and subtask dependencies is effective. Two reviewers give positive overall comments. Though Reviewer JZBq did not put efforts on discussion, the authors' feedback addressed the concern well. Reviewer 1wtR's comments are quite short thus neglected. The ACs decided to accept it.

**Justification For Why Not Higher Score:**

The model novelty is not significant while the problem is interesting and the dataset is meaningful.

**Justification For Why Not Lower Score:**

The pipeline is crucial for further AI system that combines multi-models.

---

### Decision · Program_Chairs · 2024-01-16

Accept (poster)